# LINE-1 retrotransposition in a mouse TDP-43 model of neurodegeneration marks motor cortex neurons for cell-intrinsic and cell non-autonomous programmed cell death

**Shreevidya Korada[1,2], Oliver H. Tam[3], Hunter C. Greco[1,2], Molly Gale Hammell[3,4], Josh Dubnau[1,5], Roger B. Sher[1,2]***

**1** Program in Neuroscience, Department of Neurobiology and Behavior, Stony Brook University, Stony Brook, New York, United States of America, **2** Center for Nervous System Disorders, Stony Brook University, Stony Brook, New York, United States of America, **3** Institute for Systems Genetics, NYU Langone Health, New York, New York, United States of America, **4** Department of Neuroscience & Neuroscience Institute, NYU Langone Health, New York, New York, United States of America, **5** Department of Anesthesiology, Stony Brook School of Medicine, Stony Brook, New York, United States of America

* roger.sher@stonybrook.edu

## Abstract

A key pathological feature of Amyotrophic Lateral Sclerosis (ALS) and Frontotemporal Dementia (FTD) is the loss of nuclear localization and accumulation of cytoplasmic inclusions of TAR-DNA binding protein 43 (TDP-43). TDP-43 is a nucleic acid-binding protein involved in transcriptional repression, mRNA splicing, and the regulation of retrotransposable elements (RTEs) and endogenous retroviruses (ERVs). RTEs/ERVs are mobile virus-like genetic elements that constitute about 45% of our genome and encode the capacity to replicate through an RNA intermediate and insert cDNA copies at *de novo* chromosomal locations. A causal role of RTEs/ERVs has been demonstrated in *Drosophila* in mediating both intracellular toxicity of TDP-43 and the intercellular spread of toxicity from glia to neurons. RTEs/ERVs are inappropriately expressed in postmortem tissues from ALS, FTD, and Alzheimer's Disease (AD) patients, but the role of RTEs/ERVs has not yet been examined in a vertebrate model of TDP-43 pathology. We utilized established transgenic mouse models that overexpress moderate levels of human wild-type TDP-43 or a mutant version with a specific ALS-causal Q331K amino acid substitution, together with a LINE-1-EGFP retrotransposon indicator line. We found that TDP-43 animals exhibit broad expression of RTEs/ERVs with LINE-1 retrotransposition in glia and neurons in the motor cortex. Expression begins with onset of neurological phenotypes, earlier in hTDP-43-Q331K animals and later in hTDP-43-WT. The LINE-1-EGFP retrotransposition reporter transiently labels spatially clustered groups of neurons and glia at the time of onset of motor symptoms, while EGFP-labeled neurons undergo cell death and are therefore lost over time. Unlabeled cells also die as a function of distance

**Data availability statement:** All RNAsequencing data files are available at https://www.ncbi.nlm.nih.gov/geo/query/acc.cgi?acc=GSE300423.

**Funding:** This study was funded by the National Institute of Aging (https://www.nia.nih.gov) to RBS [R01AG079898], and to JD [R01AG078788 and R01AG076493], by The ALS Ride For Life (https://alsrideforlife.org) to RBS, and by The WaterWheel Foundation (https://www.waterwheelfoundation.org/) to RBS. The funders had no role in the study design, data collection and analysis, decision to publish, or preparation of the manuscript.

**Competing interests:** The authors have declared that no competing interests exist.

from the clusters of LINE-1-EGFP labeled neurons and glial cells. Together, these findings support the hypothesis that TDP-43 pathology triggers RTE/ERV expression in the motor cortex, that such expression marks cells for programmed cell death, with cell non-autonomous effects on nearby neurons and glial cells.

## Author summary

Pathological protein aggregation marks the progression of neurodegenerative diseases (NDDs), with TDP-43 forming protein aggregates in post-mortem brains in 97% of Amyotrophic Lateral Sclerosis (ALS), 40% of Frontotemporal Dementia (FTD), and many cases of Alzheimer's and Alzheimer's Related Dementias (AD/ADRD). Retrotransposons (RTEs) and Endogenous retroviruses (ERVs) are DNA sequences derived from ancient virus-like elements that encode almost 50% of our genomes and can replicate and insert into new genomic locations, leading to cellular toxicity. Multiple silencing systems protect the genome by arresting RTE expression, but these safeguards weaken with aging, and RTEs/ERVs are broadly induced in NDDs. TDP43 was first identified as a transcriptional repressor of HIV-1, and it binds broadly to retrotransposon-derived RNA transcripts, with its dysfunction leading to inappropriate expression of RTEs. Like TDP-43, RTE alterations are seen in ALS/FTD and in AD/ADRD post-mortem brains, and in *Drosophila* models of TDP-43 pathology. Critically, the involvement of RTEs has not yet been examined during disease development and progression in a mammalian model of NDD. Here, we establish that a TDP43 mouse model recapitulates the impact of pathological TDP-43 on RTE expression, and present evidence in this dysregulation leads to both cell-intrinsic and cell non-autonomous programmed cell death.

## Introduction

ALS and FTD are two fatal neurodegenerative disorders that fall on a symptomological spectrum, which includes motor deficits, cognitive deficits, or a combination of both [1]. Approximately 10% of ALS cases are caused by mutations in any one of about thirty different genes, while 90% are sporadic [2]. Although only a small fraction of familial cases are caused by mutations in trans-active response element DNA binding protein (TDP-43), pathological misfolding and cytoplasmic aggregation of this protein is seen in 97% of ALS patients, 45% of FTD patients, and 57% of Alzheimer's Disease patients (AD) [3–5]. TDP-43 is found primarily in the nucleus, but under pathological conditions, it becomes hyperphosphorylated, loses its nuclear localization, and accumulates in cytoplasmic inclusions [6,7]. TDP-43 plays roles in a diverse set of cellular processes, including transcriptional regulation [8], pre-mRNA splicing [9], post-translational regulation [10], stress granule formation [11], and silencing of retrotransposons (RTEs) and endogenous retroviruses (ERVs) [12–22].

RTEs are virus-like elements that propagate like a retrovirus through an RNA intermediate via a "copy-and-paste" mechanism [23]. Sequences derived from RTEs constitute approximately 45% of the human genome [24]. Expression of RTEs can drive DNA damage, genomic instability, and trigger inflammatory responses. Therefore, such expression is typically suppressed through DNA methylation [25], chromatin remodeling [26,27], siRNA silencing [28] and transcriptional repression [29]. However, many of these silencing mechanisms falter with age and in the context of neurodegenerative disorders. TDP-43 protein binds extensively to RTE transcripts, and this binding is lost in patients with FTD [21]. Upregulated expression of Human Endogenous Retrovirus-K (HERV-K) has been observed to be associated with TDP-43 proteinopathy in postmortem motor cortex (MC) tissue of ALS subjects, [13,18,20,22]. A loss of TDP-43 from neuronal nuclei in cortical tissue of FTD-ALS patients is associated with decondensation of suppressive heterochromatin around the promoters of LINE element RTEs [19]. Stratification of 148 ALS patient postmortem cortex samples based on expression profiles resulted in three distinct molecular subtypes, with one representing RTE derepression combined with TDP-43 mislocalization [13,30].

As in many other biological contexts, overexpression of hTDP-43 in *Drosophila* triggers formation of TDP-43 pathology, including loss of nuclear localization and accumulation in cytoplasmic inclusions [31,32]. This results in locomotor impairment, shortened lifespan, and derepression of many RTEs, including an endogenous retrovirus called mdg4-ERV [12]. The expression of hTDP-43 in *Drosophila* glia also results in mdg4-ERV replication, DNA damage, and apoptosis within hTDP-43-expressing cells and in nearby neurons [14,15]. The cell non-autonomous effects on neurons can be rescued by knocking down mdg4-ERV within glial cells, indicating that the ERV expression also contributes to this spread of toxicity [12,14,15]. The expression of either the mdg4-ERV in fly cells or of HERV-K in human cells is also sufficient to trigger onset of TDP-43 protein pathology, suggesting that TDP-43 aggregation and ERV expression exist in a positive feedback loop [15].

To date, research examining the role of RTEs and ERVs in TDP-43 pathology has been performed primarily in *Drosophila,* human postmortem brain tissue, and in cell culture. To investigate functional contributions of RTEs in vertebrate animals, we used an established transgenic mouse model that overexpresses moderate levels of either a human wild-type TDP-43 (hTDP-43-WT) or a human TDP-43 containing the Q331K (hTDP-43-Q331K) causal amino acid substitution found in some inherited forms of disease [33]. This mouse model yields human TDP-43 overexpression that is about 1.5X the levels of the endogenous mouse TDP-43, thereby avoiding potential artifacts that may arise from expression outside of the physiological range. Although this modest over-expression does not trigger TDP-43 nuclear clearance and accumulation of pathological TDP-43 inclusions, it does cause loss of nuclear function of TDP-43, as evidenced by defects in splicing of canonical TDP-43 targets. We report that this transgenic hTDP-43-Q331K expression causes significant derepression of RTEs and ERVs in the MC preceding the onset of motor dysfunction. We also find that this is associated with an increase in LINE-1 retrotransposition events in both neurons and glia. Such LINE-1 retrotransposition events are associated with programmed cell death in both the cells where LINE-1 retrotransposition occurs and in nearby cells. These findings establish that RTE derepression and retrotransposition are tightly correlated with the effects of TDP-43-mediated neurodegeneration both in terms of neurological phenotypes as well as at the cellular level.

## Results

### hTDP-43-Q331K mice exhibit early-onset motor deficits, while hTDP-43-WT mice exhibit late-onset motor deficits

We used transgenic mouse models that have previously been described to yield modest levels of human TDP-43 (hTDP-43-WT & hTDP-43-Q331K) expression (1.5x), and gradual appearance of neuro-pathological impacts [33]. To establish a baseline of the neurological phenotypes in our hands, we first quantified hindlimb clasping and latency to fall on the accelerated rotarod (Fig 1). As has been previously reported [33], we observe an age-dependent appearance of motor deficits that are more severe and occur early in life in the hTDP-43-Q331K transgenic animals than in the hTDP-43-WT transgenic animals. The hTDP-43-Q331K animals did not show significant differences in hindlimb clasping scores compared to the

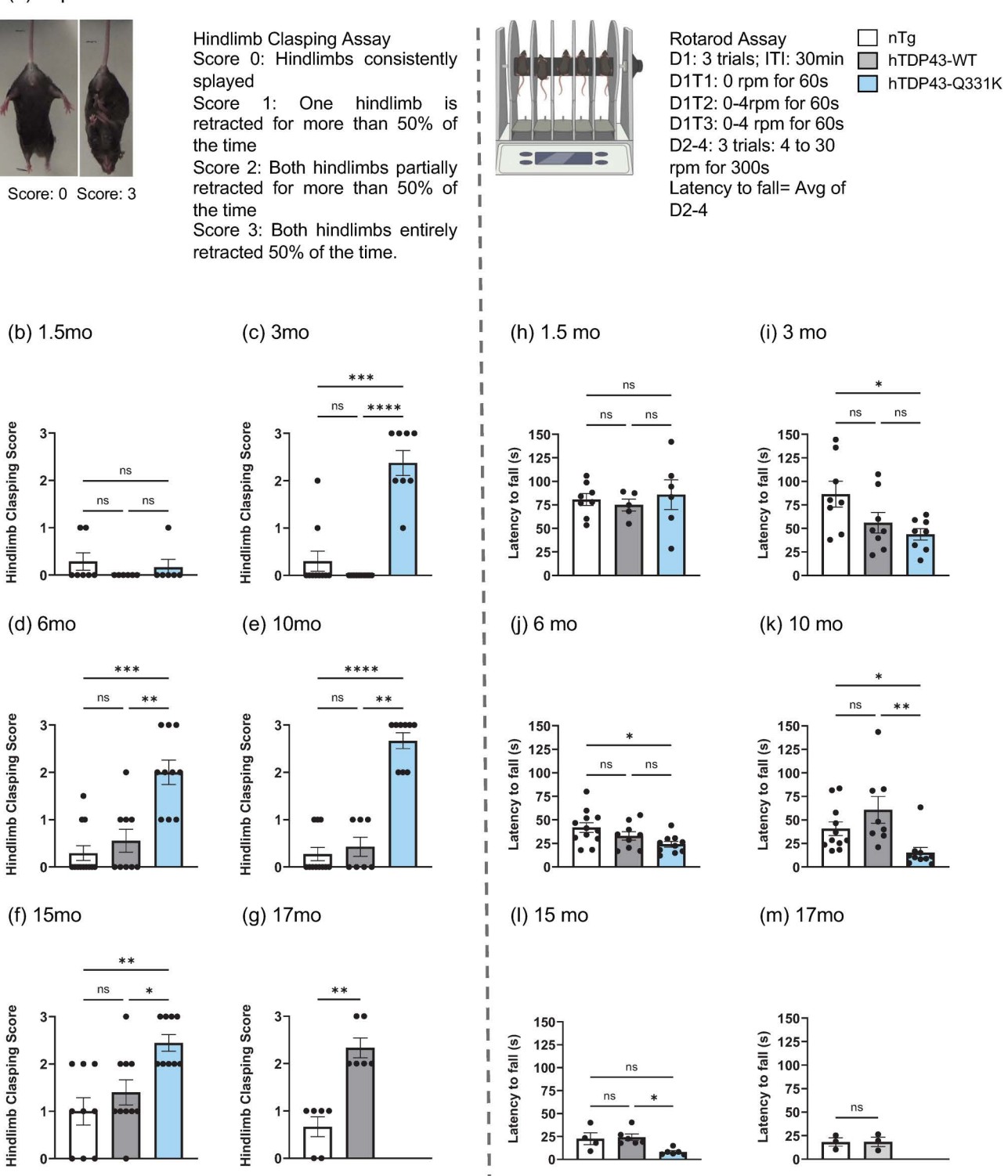

**Fig 1. hTDP-43-Q331K Tg mice start showing motor deficits at 3 months, and hTDP-43-WT Tg animals start showing deficits at 17 months.**
**(a)** Experimental outline for hindlimb clasping scoring and rotarod assay. Rotarod image created in Biorender. **(b-g)** Hindlimb Clasping at 1.5, 3, 6, 10, 15 and 17 months. Kruskal- Wallis and Dunn's multiple comparisons test was used with * p ≤ 0.05, ** p < 0.01, *** p < 0.001, **** p < 0.0001. n-=6–12

animals for each group were used. **(h-m)** Latency to fall on the accelerating rotarod at 1.5, 3, 6, 10, 15, 17 months. D1: Day 1, T1: Trial 1, ITI: Inter-trial interval. The average latency to fall (s) for days 2–4 was calculated for each animal. Normality and lognormality were tested using the D'Agostino & Pearson, Anderson-Darling, Shapiro-Wilk, and Kolmogorov-Smirnov tests. If the dataset passed all the tests, an ordinary one-way ANOVA was done with Sidak's multiple comparisons test to get the adjusted p-value. For non-normal distributions, the Kruskal- Wallis and Dunn's multiple comparisons tests were used with * $p \leq 0.05$, ** $p < 0.01$, *** $p < 0.001$, **** $p < 0.0001$. n = 5–12 animals were used for all genotypes and age groups except for the 15-month time point where n = 4–6 animals were used. Created in BioRender. Korada, S. (2025) https://BioRender.com/xatk8c9ss.

non-transgenic animals at 1.5 months (Fig 1b). However, they developed significant hindlimb clasping defects by three months of age (Fig 1c) (p = 0.0006 for the non-transgenic vs hTDP-43-Q331K comparison and p < 0.0001 for the hTDP-43-WT vs hTDP-43-Q331K comparison. Kruskal-Wallis test and Dunn's multiple comparison tests were used to calculate p values with * $p \leq 0.05$, ** $p < 0.01$, *** $p < 0.001$, **** $p < 0.0001$). This deficit persisted throughout their lifespan (Fig 1c–1f). The hTDP-43-WT mice have previously been reported to show no significant differences from the non-transgenic littermates in hindlimb clasping at 10 months of age [33]. Our findings concur with this conclusion. However, we additionally examined hindlimb clasping at 15 months and 17 months (Fig 1f–1g) and observed that the hTDP-43-WT mice showed significantly higher hindlimb clasping scores at 17 months when compared with non-transgenic littermates (p = 0.0022 using the Mann-Whitney test to calculate p values and ** representing p < 0.01). In terms of latency to fall, the hTDP-43-Q331K animals exhibited shorter latency (p = 0.0305 for the non-transgenic vs hTDP-43-Q331K comparison using ordinary one-way ANOVA) on the accelerated rotarod beginning at three months (Fig 1h and 1i), consistent with previously reported motor deficits for this genotype [33]. In contrast, the hTDP-43-WT mice did not show significant differences from the non-transgenic animals on the accelerated rotarod through 17 months (Fig 1j–1m). These results are similar to those seen in our double-heterozygous crosses to the L1-EGFP reporter mouse line (hTDP-43-Q331K/L1-EGFP, hTDP-43-WT/L1-EGFP, L1-EGFP controls) (S1 Fig; and see below). Thus, we observe early-onset motor deficits in hTDP-43-Q331K mice and late-onset deficits in hTDP-43-WT mice. These phenotypes are consistent with previous reports and further establish a late-onset motor dysfunction in the hTDP-43-WT mice [33].

## hTDP-43-Q331K and hTDP-43-WT mice exhibit broad changes in both RTE and gene transcripts in the motor cortex that correlate with the onset of neurological defects

Next, we examined changes in MC transcriptional profiles over a time course in hTDP-43-WT and hTDP-43-Q331K compared to non-transgenic control animals. We extracted total RNA from MC at 1.5, 3, 6, 10, and 15 months and performed bulk RNA sequencing using the NextSeq 500 platform. Given the literature that implicates RTEs in TDP-43-related neurodegeneration [12–15,18–22,34], we analyzed the data using a well-established computational pipeline that also permits examination of repetitive reads, such as those that derive from RTE expression [35]. The features were then computationally clustered based on Spearman correlation to group genes and RTEs with similar trends.

Overall, we observed a broad differential expression of RTEs and genes in the MC, with the most dramatic effects manifesting transiently at 3 months in the hTDP-43-Q331K and at 15 months in hTDP-43-WT animals when compared to non-transgenic littermates (S2 and S3 Figs). In each hTDP-43-Q331K and hTDP-43-WT genotype, these transient changes in expression of many cellular genes, RTEs, and ERVs occur mainly at the time when the behavioral deficits first appear, or just prior to that. These effects are clearly visualized via heatmaps that we generated using combined lists of genes and RTEs that are differentially expressed across all comparisons while accounting for age, utilizing a generalized linear model (Fig 2). Although RNA sequencing has not previously been performed for this mouse model, the differential expression identifies many cellular transcripts that have previously been seen in other models of neurodegeneration as well as in human tissues ([13,30,36–49]; S2–S13 Figs).

When examining effects on RTEs and ERVs, across the genotypes and ages, the differentially expressed elements break out into two main categories. First, there are a group of RTEs/ERVs whose expression increases with age irrespective

## (a) Significantly DE RTE in the motor cortex

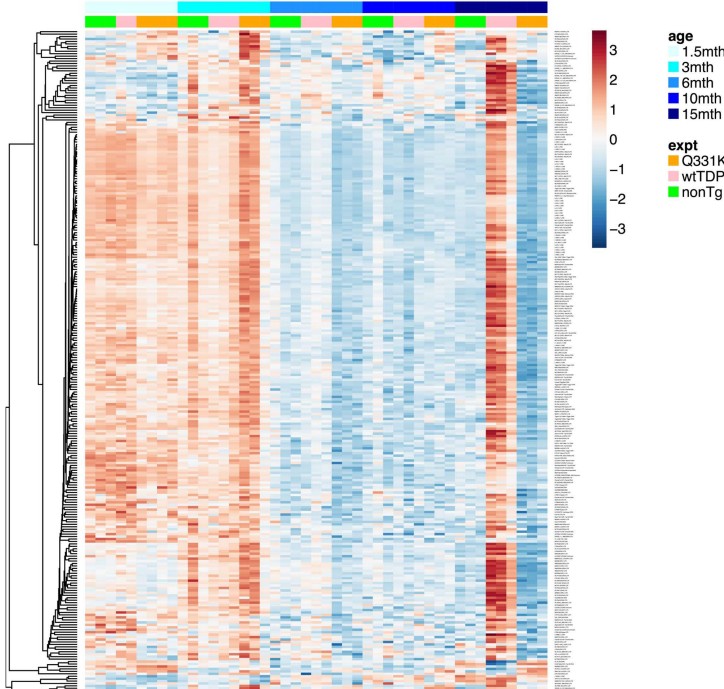

## (b) Significantly DE active mouse LINE family RTEs

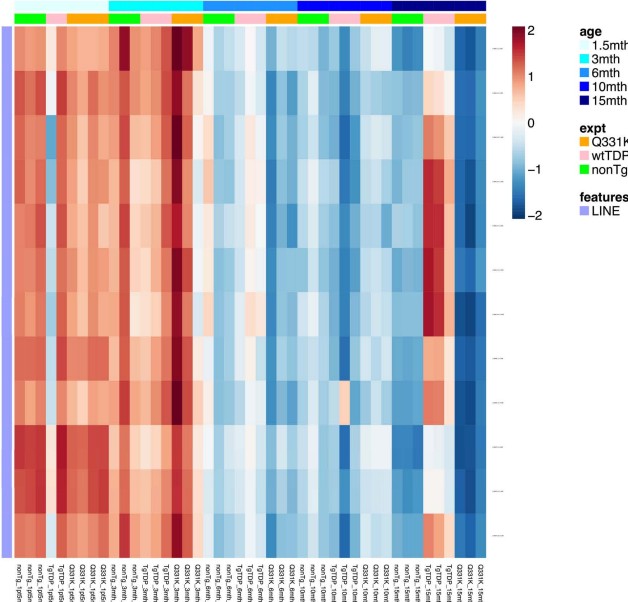

## (c) Significantly differentially expressed RTEs at different time points

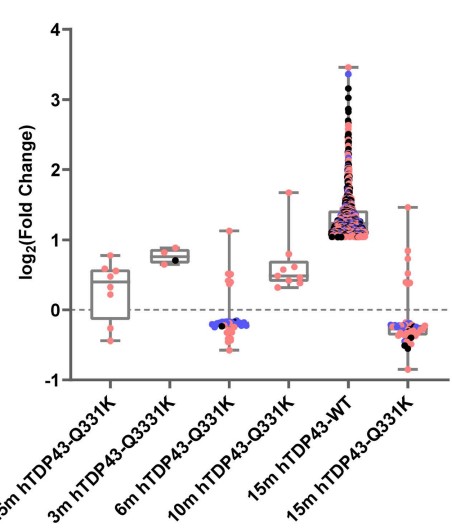

**Fig 2. hTDP-43-Q331K Tg mice show elevated levels of retrotransposable elements (RTE) at 1.5 and 3 months, and hTDP-43-WT Tg animals show elevated RTE expression at 15 months. (a)** Differentially expressed RTE across 1.5, 3, 6, 10, and 15 months in hTDP-43-Q331K Tg and hTDP-43-WT Tg mouse MC. **(b)** Active mouse LINE family RTE show significant downregulation at 6 months and 15 months and hTDP-43-WT Tg animals

show significant upregulation at 15 months n = 3 mixed-sex, age-matched cohorts were used for all genotypes and age groups except 1.5 month, where n = 2 for nTg and n = 4 for hTDP-43-WT Tg were used. See Methods for details regarding the analysis pipeline, including statistical analyses. **(c)** log2fold change values for significantly differentially expressed RTE (padj<0.05 for all RTEs on the plot). Significant differential expression of RTEs seen at all time points starting at 1.5 months for the hTDP-43-Q331K mouse but only shows up at 15 months in the hTDP-43-WT mouse MC.

of genotype (S12 Fig) ($p_{adj}$ = 2.37E-08, odds ratio = infinite). This is consistent with a prior literature that establishes age-dependent derepression of many RTEs and ERVs [34,50–55]. Second, there are a set of RTEs/ERVs that are expressed at higher levels in the context of our neurodegeneration models. This group of elements show elevated expression in MC at the 3-month age in the hTDP-43-Q331K animals and at the 15-month age in the hTDP-43-WT animals. We also note that this group of RTEs and ERVs whose expression is induced in the context of TDP-43 neurodegeneration also have relatively high expression in all three genotypes in young animals that are just 1.5 months old compared to later ages (Fig 2a; (padj = 9.92E-76, odds ratio = 13.298)). This is consistent with the possibility that this set of elements that are impacted by neurodegeneration also are expressed during development [56–58] and exhibit residual RTE transcripts early in life.

The set of RTE and ERV elements that exhibit high expression at 1.5 months of age in control animals but low expression at older ages are largely the same subset that are significantly increased in the MC at 3 months in the hTDP-43-Q331K and at 15 months in the hTDP-43-WT transgenic animals ($p_{adj}$ = 1.03E-15, odds ratio = 9.934). In the two TDP-43 transgenic groups, the most dramatic effects on both cellular transcripts (S4–S7 and S13 Figs) and RTEs/ERVs occur transiently, around the time the neurological defects emerge. For example, compared with non-transgenic littermates, the hTDP-43-Q331K animals show an overall elevation of many different RTEs, ERVs and cellular transcripts at the 3-month age (Figs 2a and S2), which correlates with the time-point when neurological defects occur in this genotype (Fig 1). Unlike the neurological effects, which progress and persist, these effects on the expression of RTEs and ERVs are transient. Although the increase in overall expression of many RTEs and ERVs at 3 months is largely transient, there are subsets of elements that show continued elevation across all timepoints (Fig 2c).

In contrast with hTDP-43-Q331K, the hTDP-43-WT animals showed no significant differences in RTE/ERV expression when compared to non-transgenic littermates at the 1.5, 3, 6, and 10-month time points. However, at 15 months, the hTDP-43-WT animals demonstrated a significant increase in 908 RTEs and ERVs (Fig 2c). Principal component analysis plots for significantly expressed RTEs and genes (S8–S10 Figs) show a clear separation of the hTDP-43-WT biological replicates from the other groups in PC1 (S8 Fig (i)). The hTDP-43-Q331K also shows a separation from the other samples with one outlier. The biological replicates for the 3-month and 15-month time points (across all genotypes) show clear separation from all the other time points on PC3 of the principal component analysis (S8 Fig (vii)).

The effects of hTDP-43-Q331K at 3 months and hTDP-43-WT at 15 months on RTE and ERV expression are quite broad, impacting each of the major clades of RTEs. Because LINE elements are derepressed in postmortem brain samples from human patients [13,19] and because a subset of LINE elements are replication competent, we next examined the effects of TDP-43 transgenes on the active subset of mouse LINE elements. We found that the overall pattern of differential expression of active mouse LINE elements matches that seen across the other types of elements: expression of active LINE elements increased in the MC of 3-month-old hTDP-43-Q331K animals and was significantly elevated at 15 months in the hTDP-43-WT animals (Fig 2b). These effects were similar across both evolutionarily recent and more ancient transposable elements (Fig 2a and 2c).

### hTDP-43-Q331K and hTDP-43-WT mice show increased numbers of cells with LINE-1 retrotransposition events in the motor cortex

LINE-1 elements have previously been shown to be derepressed in postmortem cortical neurons that exhibit mislocalization of TDP-43 in FTD subjects [19]. In our RNA sequencing experiments, we observed that active LINE-1 elements are

transiently upregulated around the time of neurological symptom onset, at 3 months in hTDP-43-Q331K and at 15 months in hTDP-43-WT (Fig 2b). Because both mouse and human genomes contain fully active copies of LINE-1 elements, their expression may result in retrotransposition. To investigate this possibility, we used an established L1-EGFP indicator cassette that expresses GFP only after the transgenic LINE reporter element has passed through an RNA intermediate and reinserted as a cDNA into a *de novo* chromosomal location [59,60]. We crossed the hTDP-43-Q331K and the hTDP-43-WT animals to the L1-EGFP reporter line to generate double-heterozygous offspring.

We performed immunohistochemistry in the MC of the hTDP-43-Q331K/L1-EGFP and hTDP-43-WT/L1-EGFP double transgenic animals, as well as their L1-EGFP littermates, to examine the presence of GFP-positive cells. For each genotype, we then imaged GFP-positive cells across a time-course. As expected from previous reports [60], we observed sparse GFP labeling in the L1-EGFP littermates of a relatively small number of neurons and glial cells in MC across all ages and genotypes (Figs 3a–3e and S14, S15). This is presumably the result of rare de novo retrotransposition of LINE-1 elements that can take place in neuronal and glial lineages during the last rounds of cell division in normal development. In the L1-EGFP animals, we see very few GFP-positive cells per section in MC across all ages. By contrast, in the hTDP-43-WT/L1-EGFP and hTDP-43-Q331K/L1-EGFP genotypes, we see spatially defined groups of many GFP-positive neurons and glia in the MC. Importantly, this effect has a stochastic onset, appearing in very few animals at 1.5 months of age, but in a higher fraction of animals at the 3- and 6-month timepoints (Fig 3a–3c and S1 Table). Because these large, spatially defined groups of L1-EGFP-labeled cells are rarely seen in the hTDP-43-Q331K/L1-EGFP and the hTDP-43-WT/L1-EGFP genotypes at 1.5 months of age but are more frequently detected in 3- and 6-month-old animals, we presume that they are due to independent retrotransposition events taking place after development. Such large groups of GFP-labeled cells are not observed in the L1-EGFP single transgenic genotype at any age.

But the presence of such large groups of labelled cells is a transient effect that is no longer detected at 10- and 15-months (Fig 3d and 3e and S1 Table). For example, at 3 months, the hTDP-43-Q331K/L1-EGFP animals exhibited significantly higher numbers of GFP-positive cells per section compared to their L1-EGFP littermate controls (C2 control in Fig 3b; $p = 0.0406$. Box-Cox transformation was done with a $\lambda = 0.041$ followed by One-Way ANOVA and student t-tests post hoc to compute values with * $p \leq 0.05$, ** $p < 0.01$, *** $p < 0.001$, **** $p < 0.0001$) (Fig 3b). The hTDP-43-WT/L1-EGFP animals also displayed significantly higher GFP-counts per section than the L1-EGFP littermate controls (C1 control in Fig 3; $p = 0.0216$) per section but with fewer cells compared to that seen in the hTDP-43-Q331K/L1-EGFP animals. The hTDP-43-WT/L1-EGFP exhibit relatively lower subject-to-subject variability in numbers of GFP expressing cells than the hTDP-43-Q331K/L1-EGFP animals, with a normal distribution (Std Dev = 5.740 for the hTDP-43-WT vs Std Dev = 11.88 for the hTDP-43-Q331K cohorts) (Fig 3b).

This effect on GFP-positive cells per section was also seen at 6 months age for both the hTDP-43-Q331K/L1-EGFP and the hTDP-43-WT/L1-EGFP animals, with significantly higher numbers of GFP-positive cells per section ($p = 0.0157$ in the hTDP-43-Q331K/L1-EGFP vs L1-EGFP littermates (C1) and $p = 0.0410$ for the hTDP-43-WT/L1-EGFP vs L1-EGFP-littermate comparison; C2) in the MC (Fig 3c). Notably, at 10 and 15 months, the number of GFP-positive cell counts per section returned to control levels (Fig 3d, 3e). Thus, both the hTDP-43-Q331K and the hTDP-43-WT animals exhibit higher numbers of cells expressing the L1-EGFP retrotransposition reporter in the MC at 3 and 6 months, but the presence of L1-EGFP-labeled cells becomes undetectable at later ages. The hTDP-43-Q331K/L1-EGFP animals exhibit these LINE-1 retrotransposition events in the MC at a time that correlates with the onset of motor dysfunction and closely follows the timing of the derepression of RTEs seen in the RNA sequencing experiments (Figs 1 and 2). In contrast, the hTDP-43-WT/L1-EGFP animals display LINE-1 retrotransposition events that occur much earlier than the onset of motor dysfunction (Figs 1 and 3b) and prior to when we observe RTE derepression in bulk tissue RNA sequencing (Fig 2). In addition to the MC, we also quantified GFP-positive cells in the striatum, nucleus accumbens, and agranular insular cortex (S14 and S16–S18 Figs). Significant increases in number of cells that exhibited LINE-1 retrotransposition events were observed in all these regions at 3 or 6 months in both genotypes (S15 Fig).

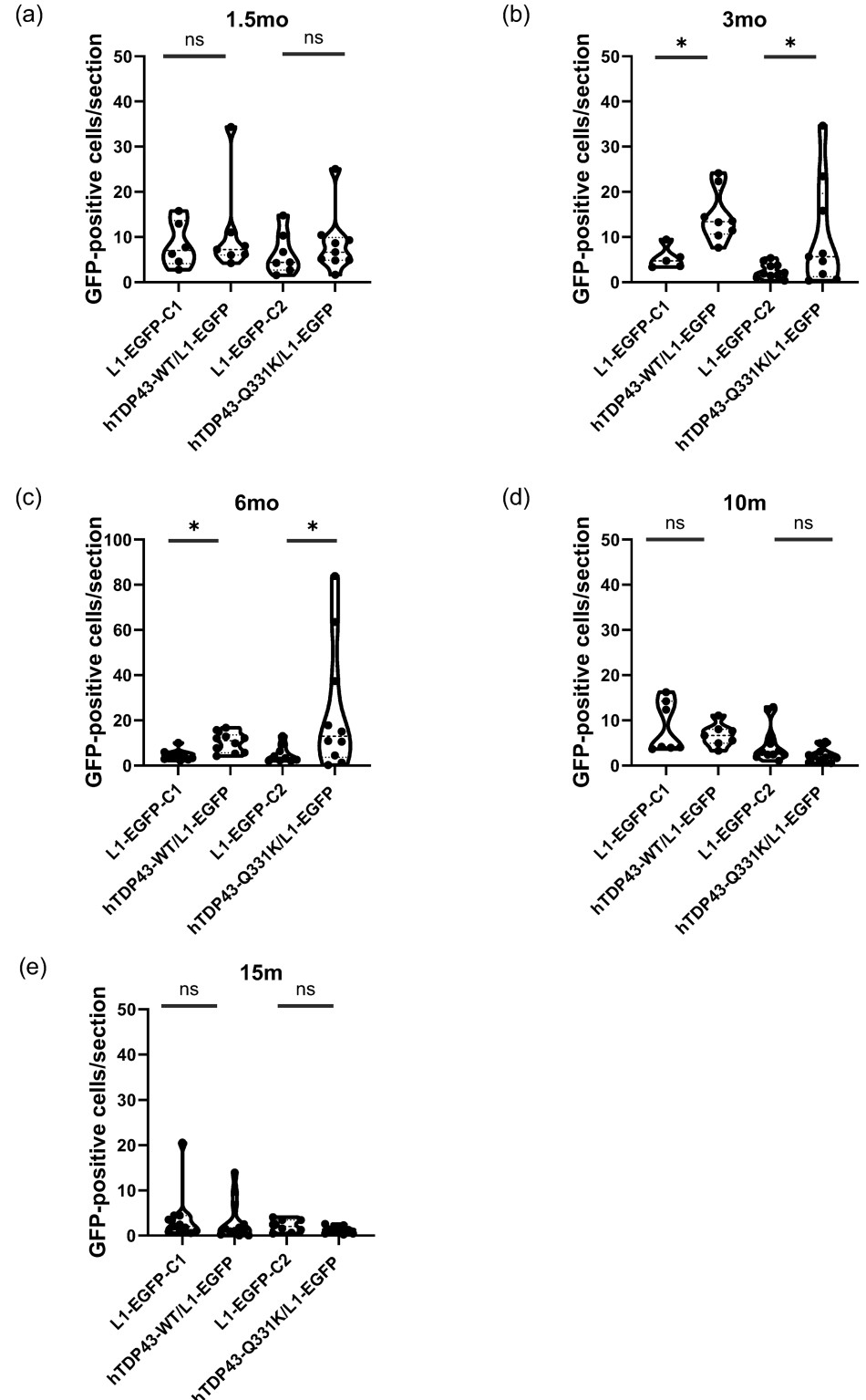

**Fig 3. LINE-1 retrotransposition events occur in cells in the MC of hTDP-43-Q331K Tg mice and hTDP-43-WT Tg mice at 3 and 6 months. (a-e)** GFP-positive cells/ section in 1.5 m, 3m, 6m, 10m, and 15m MC, respectively. Images were processed on FIJI- ImageJ and manually counted. Statistical analyses were done on JMP Statistical software. GFP counts per section were first normalized using Box-Cox transformation with a λ = 0.041. ANOVA

was used to test the effect of age and genotype on the GFP counts, and student t-tests were conducted post-hoc to compute the significance of planned comparisons with * p ≤ 0.05 and ** p < 0.01. p-value for (b) was 0.0406 and 0.0216, respectively. P-value for (c) was 0.0157 and 0.0410, respectively. Mixed cohorts of n = 5 to 11 animals were used for all genotypes and age groups. hTDP-43-WT/L1-EGFP mice and hTDP-43-Q331K/L1-EGFP mice were maintained as separate colonies and hence have been compared to their littermates L1-EGFP-C1 and L1-EGFP-C2, respectively.

Although we observe an increase in the number of GFP-labeled cells in both the hTDP-43-Q331K/L1-EGFP and hTDP-43-WT/L1-EGFP animals at 3 and 6 months of age, we noted that the distribution of labeled GFP cells along the anteroposterior axis of the MC was quite different between the two genotypes. For example, at the 6-month age (Fig 4), the L1-EGFP-labeled cells in hTDP-43-WT/L1-EGFP animals are primarily restricted to the anterior portion of the MC. But in the hTDP-43-Q331K/L1-EGFP animals, we observe large groups of GFP-positive cells with a continuous anteroposterior distribution. We also noted that the GFP label in the more posterior sections of the hTDP-43-Q331K/L1-EGFP group included many cells with morphology characteristic of pyramidal neurons (Fig 4), whereas the more anterior sections appeared to contain a mix of cells with morphology indicative of both neurons and astrocytes (see below).

### hTDP-43-Q331K animals exhibit increased numbers of LINE-1 retrotransposition events in MC neurons and astrocytes, but not microglia

Having established that the hTDP-43-Q331K/L1-EGFP and the hTDP-43-WT/L1-EGFP animals exhibit increased numbers of cells with LINE-1 retrotransposition events in the MC, we next investigated if there were cell-type-specific differences in LINE-1 retrotransposition within the spatially clustered L1-EGFP expressing cells in the two double heterozygous lines. In order to quantify cell types within these regions, we defined a cluster of cells within a given brain region as a group of GFP-labeled cells whose count was three standard deviations higher than the mean GFP-labeled cell counts of that region in the L1-EGFP control. Although multiple brain regions exhibited GFP-positive cells (Figs 3 and S14, S15), the MC was the only region that exhibited such defined clusters.

To determine the identity of labeled cells within each cluster, we performed immunohistochemistry with EGFP together with either the pan-neuronal marker Neurotrace-Blue, the astrocyte marker S100β, or the microglial marker Iba1. We observed in both hTDP-43-Q331K/L1-EGFP and hTDP-43-WT/L1-EGFP genotypes that at 3 and 6 months, all MC GFP-positive cells were labeled with either Neurotrace-Blue or S100β (Fig 5a and 5b), indicating that the cell types labeled by the LINE-1 reporter are neurons and astrocytes. We did not find any GFP-positive cells labeled with Iba1 in the MC of any of the animals, indicating that microglia are not likely a site of L1-EGFP retrotransposition.

We also quantified the total number of GFP-positive cells per cluster. We found no significant difference in the numbers of L1-EGFP-positive neurons per cluster between the hTDP-43-WT/L1-EGFP and the hTDP-43-Q331K/L1-EGFP animals at 3 months. However, at 6 months, there are significantly more L1-EGFP-positive neurons per cluster in the hTDP-43-Q331K/L1-EGFP animals compared to the hTDP-43-WT/L1-EGPP animals (p = 0.0027 using ordinary one-way ANOVA with Sidak's multiple comparisons test). We also observed that the number of LI-EGFP-positive neurons per cluster in the MC of the 6-month-old hTDP-43-Q331K/L1-EGFP animals was also significantly higher than that observed in both genotypes at 3 months (p = 0.0068 for the 6-monthhTDP-43-Q331K/L1-EGFP and 3-monthhTDP-43-WT/L1-EGFP comparison and p = 0.0128 for 6-monthhTDP-43-Q331K/L1-EGFP and 3-month hTDP-43-Q331K/L1-EGFP comparison) (Fig 5c). In contrast to the hTDP-43-Q331K/L1-EGFP animals, which show an increase in numbers of GFP-labeled neurons per cluster between 3 and 6 months, the hTDP-43-WT/L1-EGFP animals show no significant differences between those ages (Fig 5c). There were no significant differences in the number of GFP-positive astrocytes per cluster between 3 and 6 months for either genotype (Fig 5d).

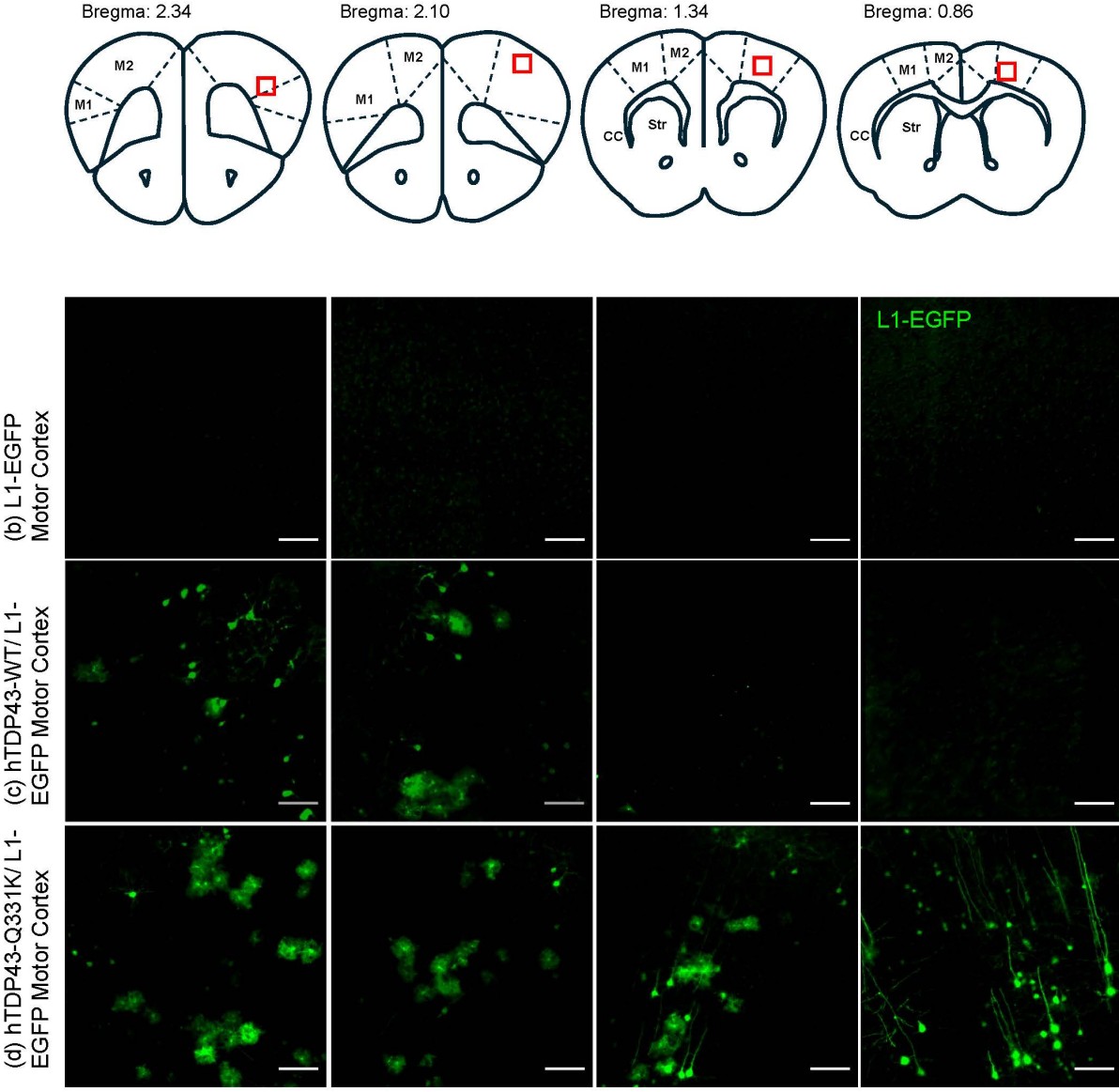

(a) LINE1 replicates in different cell types in the 6-month-old hTDP43- WT/L1-EGFP and hTDP43- Q331K/L1-EGFP motor cortex

Key:

M1- Primary Motor Cortex
M2- Secondary Motor Cortex
CC- Corpus Callosum
Str- Striatum

**Fig 4. hTDP-43-Q331K Tg and hTDP-43-WT Tg mice show LINE-1 retrotransposition events in spatially distinct groups of cells in the MC at 6 months. (a)** The Bregma coordinates (created in Powerpoint) of the brain sections in the three genotypes are shown. **(b)** Representative sections from one L1-EGFP mouse MC. **(c)** Representative sections from one hTDP-43-WT/ L1-EGFP mouse MC. **(d)** Representative sections from one hTDP-43-Q331K/L1-EGFP mouse MC. Scalebar = 50um.

## (a) 6m hTDP43-Q331K/L1-EGFP MC

## (b) 6m hTDP43-WT/L1-EGFP MC

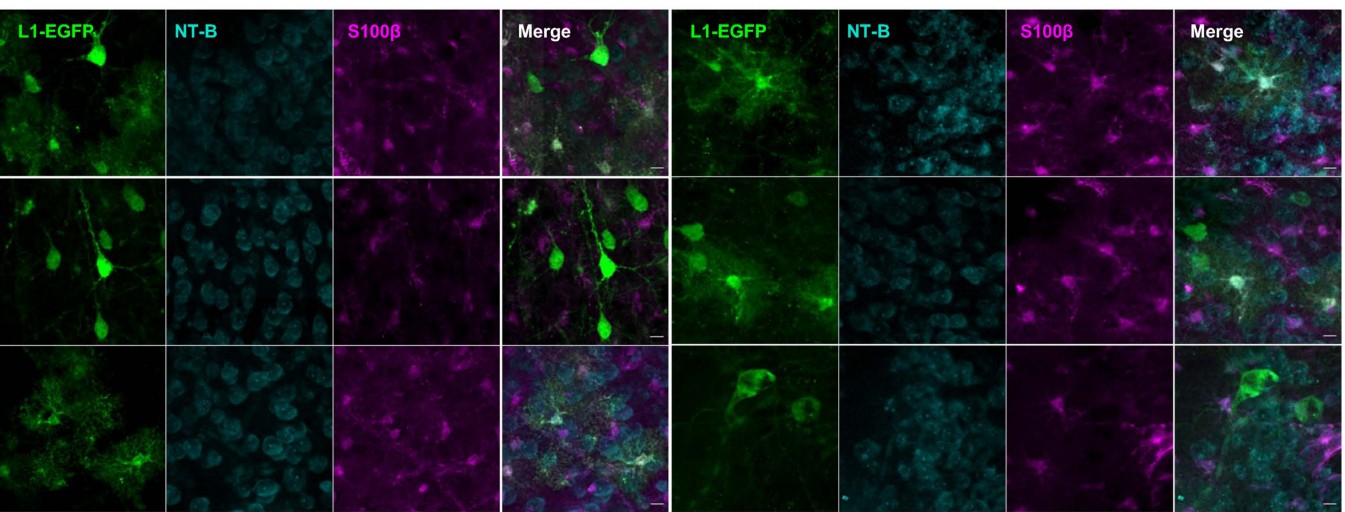

## (c) Number of GFP-positive neurons in a cluster

## (d) Number of GFP-positive astrocytes in a cluster

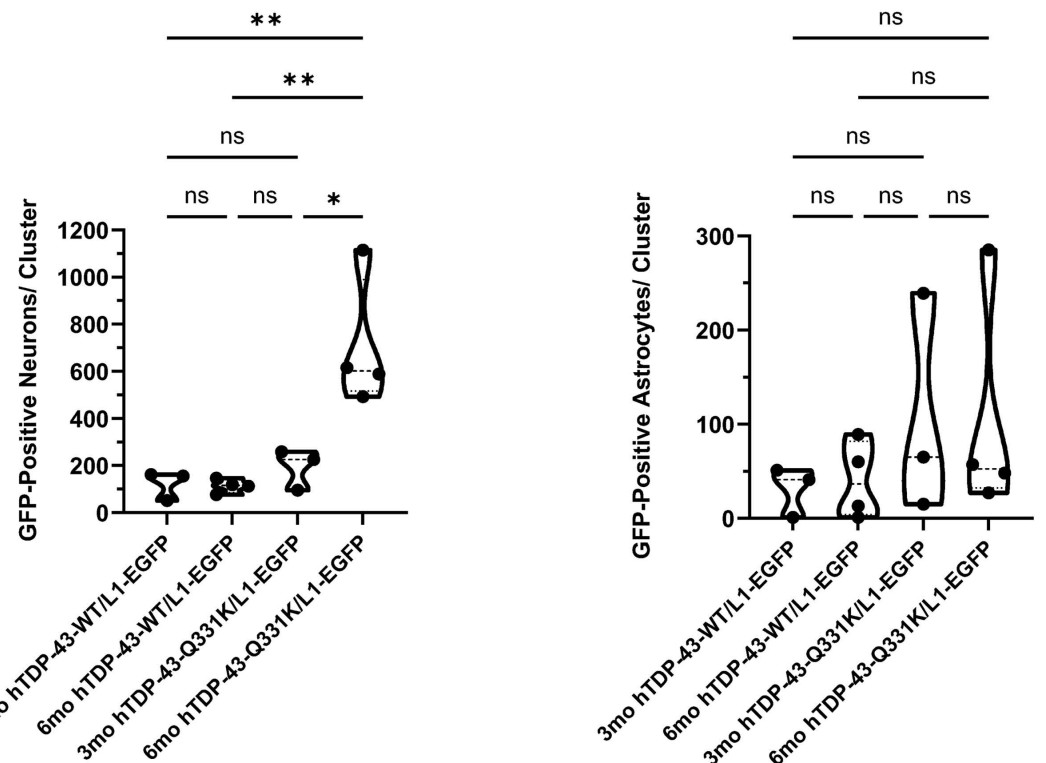

**Fig 5. LINE-1 replicates in neurons and astrocytes in the MC of hTDP-43-Q331K/ L1-EGFP and hTDP-43-WT/L1-EGFP transgenic animals. (a)** Representative 40x images of GFP-positive cells within a MC cluster in a 6-month hTDP-43-Q331K/L1-EGFP animal showing GFP, Neurotrace Blue, and S100B staining. **(b)** Representative 40x images of GFP-positive cells within a MC cluster in a 6-month hTDP-43-WT/L1-EGFP animal. Neuron **(c)** and astrocyte **(d)** counts in the 3-month and 6-month mouse MC cluster in hTDP-43-Q331K/L1-EGFP and hTDP-43-WT/L1-EGFP transgenic animals.

Cell-type-specific counts within a cluster were taken. One-way ANOVA with Sidak's multiple comparison test was used to test significance for (c), and the Kruskal–Wallis test was used for (d) with * p ≤ 0.05 and ** p < 0.01. p-value for (c) was 0.0068 for the 6-month hTDP-43-Q331K/L1-EGFP and 3-month hTDP-43-WT/L1-EGFP comparison, p = 0.0027 for the 6-month hTDP-43-Q331K/L1-EGFP and 6-month hTDP-43-WT/L1-EGFP comparison and p = 0.0128 for 6-month hTDP-43-Q331K/L1-EGFP and 3-month hTDP-43-Q331K/L1-EGFP comparison.

## hTDP-43-Q331K animals exhibit contiguous and unilateral clusters of L1-EGFP-labeled neurons that span the anterior-posterior extent of the MC

We next examined the spatial distribution of the clusters of cells that exhibited LINE-1 retrotransposition events across the anteroposterior axis in MC and compared the distribution across brain hemispheres. To do so, we examined the organization of GFP-positive clusters in 14 sections at identical bregma locations in hTDP-43-Q331K/L1-EGFP and hTDP-43-WT/L1-EGFP animals. Each section was 50 microns thick, with a 150-micron interval between sections. We adhered to the cluster definition used previously to determine the presence or absence of clusters in the MC. We first determined if an animal had a defined cluster. In animals that passed this total cell count cluster threshold, we determined if subsequent anterior-to-posterior sections contained defined clusters and if they appeared in the ipsilateral or contralateral MC from the previous section cluster (Fig 6a. pseudo-left, RED or pseudo-right, GREEN designator, see methods). Section-wise cluster thresholds were based on the L1-EGFP mean+3SD for that specific bregma anterior-posterior coordinate from L1-EGFP littermate controls. The sections within each animal that did not pass the section-wise cluster are marked as yellow (Fig 6a, see legend).

At 3 and 6 months, the hTDP-43-Q331K/L1-EGFP mice typically displayed a single large unilateral cluster that spanned multiple sections along the anteroposterior axis of the MC. In contrast, the hTDP-43-WT/L1-EGFP mice typically exhibited multiple smaller clusters that were restricted to just a few sections. These animals often exhibited small clusters on both sides of the brain within a given section (Fig 6a). The distribution of these clusters of L1-EGFP-labeled neurons also differed over time. At 3 months of age, the hTDP-43-Q331K/L1-EGFP genotype exhibited clusters of L1-EGFP labeling, typically in the anterior portion of the MC. At 6 months of age, however, clusters of neurons that are positive for the L1-EGFP reporter are more uniformly distributed along the anteroposterior axis in the hTDP-43-Q331K/L1-EGFP animals. This was not the case with the hTDP-43-WT/L1-EGFP animals, which continued to exhibit smaller clusters of L1-EGFP labeled cells, often restricted to the anterior region, with no significant changes in size or distribution between 3 and 6 months (Fig 6b). We identified significant effects of bregma coordinates, age, and genotype (p < 0.0001, as determined by a two-way ANOVA mixed-effects model on log-transformed data) on the spatial distribution of clusters of L1-EGFP-labeled cells in the MC of the mutant and wild-type forms of TDP-43. Together, the findings are consistent with the idea that the majority of cells exhibiting L1-EGFP retrotransposition in the hTDP-43-Q331K/L1-EGFP animals are part of a single, spatially contiguous cluster of cells. This cluster of cells typically appears first in the anterior regions of the MC but extends to the posterior regions by 6 months of age. By contrast, the distribution of L1-EGFP-labeled cells in the hTDP-43-WT/L1-EGFP animals indicates the presence smaller clusters of cells, which likely occurred independently at different locations along the anterior-posterior axis.

## TDP-43-driven LINE-1 retrotransposition in neurons predicts cell-intrinsic death, as well as death of cells at a distance

Both the RNA sequencing and L1-EGFP reporter studies described above indicate that RTE expression and L1-EGFP retrotransposition occur transiently around the onset of motor deficits. Although the motor symptoms persist into later ages, the L1-EGFP reporter-labeled cells appear in large clusters in the MC around the time of motor symptom onset and then are no longer detected at later timepoints. The transient nature of the presence of EGFP-labeled cells was unexpected because activation of the L1-EGFP reporter depends on retrotransposition through an RNA intermediate and insertion of a de novo cDNA copy in which a constitutive CMV promoter is fused to the EGFP cassette.

## (a) GFP cluster lateralization in the mouse motor cortex

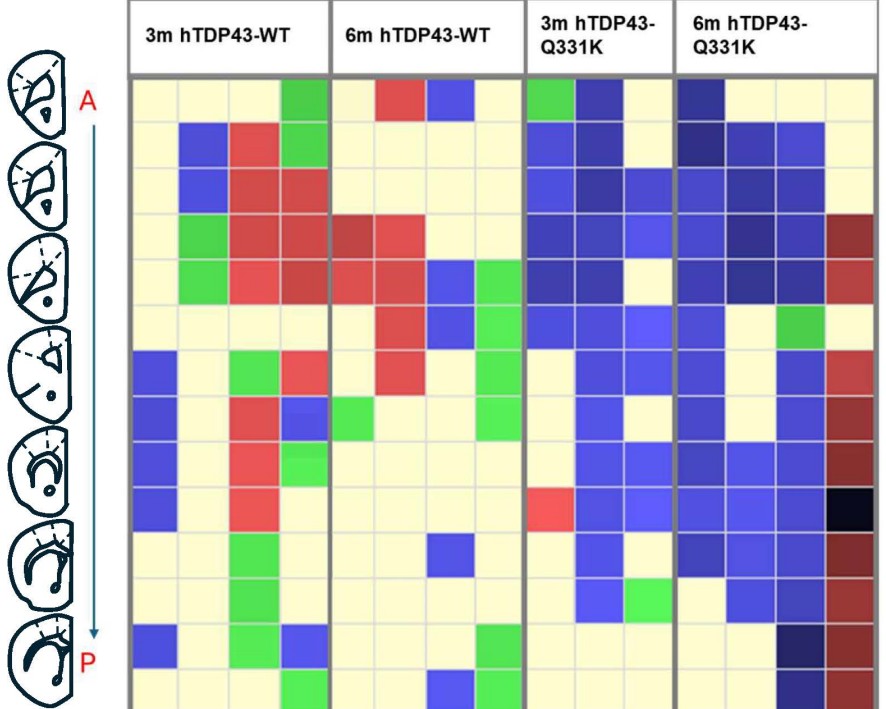

## (b) Cluster size in the TDP43-Q331K /L1-EGFP and TDP43-WT/L1-EGFP MC across the antero- posterior axis

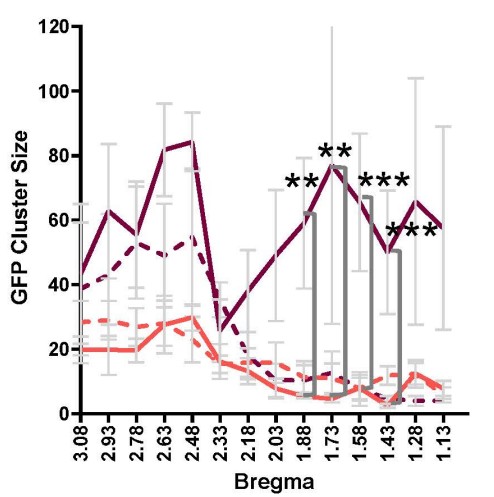

— 6mo TDP43-Q331K/ L1-EGFP

‑ ‑ ‑ 3mo TDP43-Q331K/ L1-EGFP

‑ ‑ ‑ 3mo TDP43-WT/ L1-EGFP

— 6mo TDP43-WT/ L1-EGFP

**Fig 6. LINE 1 retrotransposition occurs in GFP positive clusters that are large and unilateral in the hTDP-43-Q331K/L1-EGFP animal and small and bilateral in the hTDP-43-WT/L1-EGFP animals. (a)** Heatmap showing GFP positive cell clusters that are on the pseudo-left (labeled as left in the figure legend), pseudo-right (labeled as right in the figure legend) hemisphere, or are bilateral. The intensity of the color represents the size of the

cluster. Rows show animals, and columns show sections with bregma indicated above them. Bregma images created in Powerpoint. The absence of a cluster is indicated by pale yellow. **(b)** Size distribution across the sections in the four groups. Cluster is defined as three standard deviations from the mean of the control L1-EGFP animals calculated separately for each group. Error bars represent the standard error of the mean. p-value was computed using a two-way ANOVA on the log-transformed data with a mixed effects model and Tukey's multiple comparison test post-hoc. p=0.0066 between 6m hTDP-43-WT/L1-EGFP and 6m hTDP-43-Q331K/L1-EGFP at bregma=1.88, p=0.0088 between 3m hTDP-43-WT/L1-EGFP and 6m hTDP-43-Q331K/L1-EGFP at bregma=1.88, p=0.0023 between 6m hTDP-43-WT/L1-EGFP and 6m hTDP-43-Q331K/L1-EGFP at bregma=1.73, p=0.0336 between 3m hTDP-43-WT/L1-EGFP and 6m hTDP-43-Q331K/L1-EGFP at bregma=1.73, p=0.0009 between 6m hTDP-43-WT/L1-EGFP and 6m hTDP-43-Q331K/L1-EGFP at bregma=1.58, p=0.0011 between 3m hTDP-43-WT/L1-EGFP and 6m hTDP-43-Q331K/L1-EGFP at bregma=1.58, p=0.0051 between 6m hTDP-43-Q331K/L1-EGFP and 3m hTDP-43-Q331K/L1-EGFP at bregma=1.58, p=0.0412 between 6m hTDP-43-WT/L1-EGFP and 3m hTDP-43-WT/L1-EGFP at bregma=1.43, p=0.0002 between 6m hTDP-43-WT/L1-EGFP and 6m hTDP-43-Q331K/L1-EGFP at bregma=1.43, and p=0.0334between 6m hTDP-43-Q331/L1-EGFP and 3m hTDP-43-Q331K/L1-EGFP at bregma=1.43.

One possible explanation for the absence of L1-EGFP-labeled cell clusters at later timepoints is the death of the neurons and glia in which LINE-1 retrotransposition events take place. To test this possibility, we performed immuno-histochemistry using the TUNEL assay in conjunction with DAPI, to label all nuclei, and GFP to label neurons or glia in which the LINE-1 indicator cassette reveals that a retrotransposition event has occurred. We found that L1-EGFP-positive neurons were often positive for TUNEL. This was true in all three genotypes, regardless of the presence of the TDP-43 wild-type or mutant transgenes (Fig 7a). 75% of GFP-positive neurons in L1-EGFP, 79% of those in hTDP-43-Q331K/L1-EGFP mice and 52% in hTDP-43-WT/L1-EGFP mice were positive for TUNEL (Fig 7c). Thus, neurons in which the L1-EGFP reporter indicates a retrotransposition event has taken place are highly likely to undergo programmed cell death, irrespective of whether the animal contains the human TDP-43 transgene. In contrast, GFP-positive astrocytes with LINE-1 retrotransposition events did not exhibit TUNEL staining, suggesting a cell-type-specific effect of LINE-1 retrotransposition-associated programmed cell death (Fig 7b). But while the correlation between LINE-1 retrotransposition and TUNEL label in neurons did not differ across genotypes, the numbers of such neurons that are labeled by the L1-EGFP reporter are far higher in the hTDP-43-Q331K animals (Figs 3–6).

We also noticed that in the hTDP-43-Q331K/L1-EGFP animals, there was TUNEL labeling in many cells that were not labeled with GFP. This suggested the possibility that cells that exhibit LINE-1 retrotransposition events driven by hTDP-43-Q331K may exhibit cell non-autonomous effects leading to death of nearby neurons or glia. We therefore quantified TUNEL labeling in cells that were GFP-negative as a function of distance from clusters of L1-EGFP-positive cells. To accomplish this, we tiled confocal images over a 300μmX900μm area of MC tissue, beginning near the edge of positive clusters of L1-EGFP-labeled cells in a hTDP-43-Q331K/L1-EGFP and the hTDP-43-WT/L1-EGFP animals. We then quantified the TUNEL labeling within 100μmX100μm bins, allowing us to calculate the percentage of GFP-negative nuclei that were positive for TUNEL in each bin, as a function of distance from the edge of the cluster of GFP-labeled cells (Fig 7d).

In the L1-EGFP control mice, the GFP-negative cells located in bins closer to the rare GFP-positive cells showed no significant difference in TUNEL staining when compared to the bins that were further away (Fig 7e). Thus, in animals that do not express hTDP-43-Q331K or hTDP-43-WT, the percentage of nuclei undergoing programmed cell death was similar in regions that were near or far from the cells with LINE-1 retrotransposition. By contrast, in the hTDP-43-Q331K/L1-EGFP and hTDP-43-WT/L1-EGFP mice, we observed many GFP-negative cells labeled with TUNEL near GFP clusters, with the percentage of TUNEL-positive nuclei decreasing as a function of distance from the GFP-positive cell cluster (Fig 7e). We found that both the TDP-43 genotype (p<0.0001) and the distance (p=0.0008) from the L1-EGFP retrotransposition cluster had significant effects on the percentage of nuclei that were TUNEL-positive. Both hTDP-43-Q331K/L1-EGFP and hTPD43-WT/L1-EGFP showed significantly higher numbers of TUNEL-positive, GFP-negative nuclei compared to L1-EGFP controls at a distance from L1-EGFP clusters, with hTDP-43-Q331K/L1-EGFP having significantly higher numbers of TUNEL labeled cells than hTDP-43-WT/L1-EGFP or L1-EGFP controls at each distance up to 700 μm (Fig 7e). Beyond 700 μm, we did not observe significant differences in the number of cells undergoing programmed cell death

(a) L1- EGFP-positive neurons show TUNEL staining in all genotypes

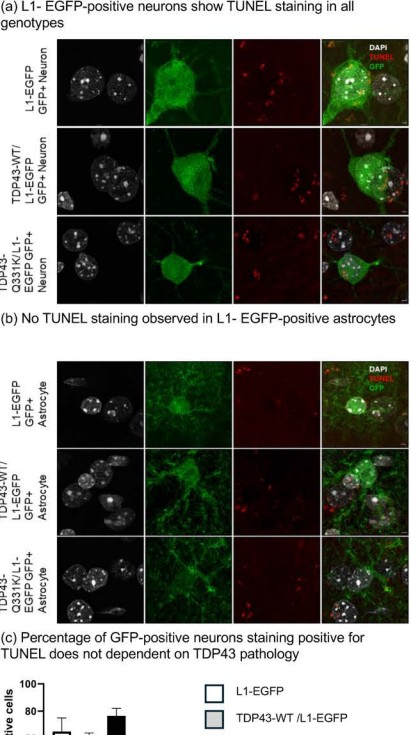

(b) No TUNEL staining observed in L1- EGFP-positive astrocytes

(c) Percentage of GFP-positive neurons staining positive for TUNEL does not dependent on TDP43 pathology

- L1-EGFP
- TDP43-WT /L1-EGFP
- TDP43-Q331K /L1-EGFP

(d) Percentage of nuclei staining positive for TUNEL reduces with distance from the GFP Cluster

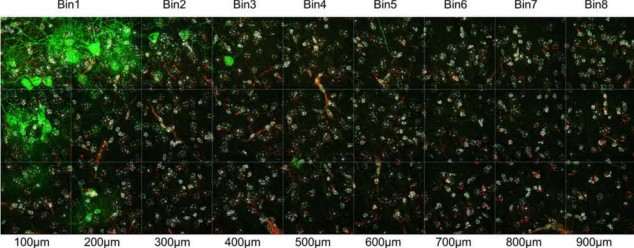

(e)

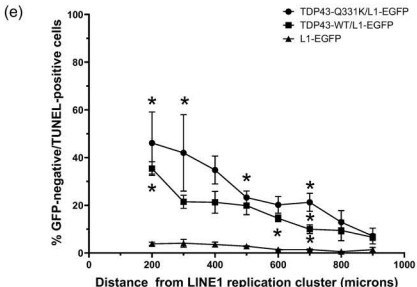

- TDP43-Q331K/L1-EGFP
- TDP43-WT/L1-EGFP
- L1-EGFP

**Fig 7. Most neurons with LINE-1 retrotransposition undergo programmed cell death, regardless of TDP-43 pathology.** But in the presence of TDP-43 pathology, cells closer to the GFP-positive cells are more likely to undergo cell death. **(a)** Representative images of GFP-positive neurons within a cluster taken at 63X with Airyscan in the 6-month MC of the hTDP-43-WT/L1-EGFP and hTDP-43-Q331K/L1-EGFP and cortex of L1-EGFP littermates.

IHC done for TUNEL, DAPI and GFP. **(b)** Representative images of GFP-positive astrocytes within a cluster taken at 63X with Airyscan in the 6-month MC of the hTDP-43-WT/L1-EGFP and hTDP-43-Q331K/L1-EGFP and L1-EGFP littermates. **(c)** Probability of GFP-positive neurons staining for TUNEL in the 6-month L1-EGFP cortex and the hTDP-43-WT/L1-EGFP and hTDP-43-Q331K/L1-EGFP mouse MC. **(d)** Percentage of nuclei staining positive for TUNEL at different distances from the edge of the GFP cluster in the hTDP-43-WT/L1-EGFP and hTDP-43-Q331K/L1-EGFP mouse MC and from a GFP-positive cell anywhere in the L1-EGFP mouse cortex. **(e)** Quantification of the percent of TUNEL-positive nuclei at different distances from the cell cluster with LINE-1 retrotransposition cluster. At 200 µm, hTDP-43-Q331K/L1-EGFP vs L1-EGFP littermates: adjusted p = 0.0341 using two-way ANOVA with Tukey's multiple comparison test; at 300 µm and 500 µm hTDP-43-Q331K/L1-EGFP vs L1-EGFP littermates: adjusted p = 0.0211 and p = 0.0213, respectively; at 600 µm hTDP-43-WT/L1-EGFP vs L1-EGFP littermates: adjusted p = 0.0211; at 700 µm hTDP-43-Q331K/L1-EGFP vs hTDP-43-WT/L1-EGFP and L1-EGFP littermates: adjusted p = 0.0211 and p = 0.0211, respectively; at 700 µm hTDP-43-WT/L1-EGFP vs L1-EGFP littermates: adjusted p = 0.0211. Scalebar = 2 µm.

across the three genotypes. Thus, we demonstrate that L1-EGFP-negative cells that are near clusters of L1-EGFP-labeled cells also undergo programmed cell death, and that the likelihood of such cell death falls off as a function of distance from the cluster of cells in which the L1-EGFP reporter has replicated.

## Discussion

Previous work in a fly model, in cell culture and in postmortem tissues from ALS and FTD subjects has consistently established that many RTEs and ERVs exhibit abnormally high expression when TDP-43 function is disrupted [12,13,16–22,30]. In flies, where genetic manipulations are more facile, it has also been possible to establish that ERV expression contributes causally to the toxicity from TDP-43 pathology [12,14,15]. But until now, the role of RTEs and ERVs has not been investigated in vivo in a vertebrate model. We establish that a mammalian animal model recapitulates the impact of pathological TDP-43 expression on RTE and ERV expression. To accomplish this, we used a model that expresses a modest level of overexpression of either the wild-type or the Q331K mutant form of hTDP-43 [33]. Such modest levels of over-expression avoid potential artifacts that may arise with higher levels of transgenic expression. While this model does not trigger the formation of cytoplasmic inclusions, it does lead to a loss of nuclear function of TDP-43, as evidenced by the disruptions seen in the splicing of established TDP-43 targets [33]. Our findings establish that transgenic expression of either the wild-type or the Q331K mutant protein is sufficient to trigger widespread, but transient derepression of numerous RTEs and retrotransposition of a well-established human LINE-1 reporter cassette in the MC. The activation of this element could reflect promoter specific transcriptional effects. But we favor the idea that instead, it reflects a more general loss of heterochromatin, as previously discussed.

Importantly, the effects on RTE expression seen with bulk-RNAseq and with the LINE-1 retrotransposition reporter begin at 3 months age in the hTDP43-Q331K animals, which corresponds well with when motor dysfunction first appears. In hTDP-43-WT animals, expression of RTEs as measured by RNAseq begins at older age that also matches when motor dysfunction begins in that genotype. But expression from the L1-EGFP retrotransposition reporter is seen earlier in these animals, at about the same age as in the hTDP43-Q331K group. The motor dysfunction is persistent and progressive, the broad expression of RTEs in MC is transient, and the presence of cells labeled with the L1-EGFP retrotransposition reporter is ephemeral. The transience of this effect appears to be caused by the cell death of the neurons that express L1-EGFP retrotransposition reporter. We also observe the death of many cells in MC that are not labeled by the LINE-1 reporter, with this cell non-autonomous effect falling off as a function of distance from the cluster of L1-EGFP labeled cells. Importantly, the cell-intrinsic effects that lead to TUNEL labeling are seen in L1-EGFP labeled neurons irrespective of whether the animals contain a hTDP-43 transgene. This effect is not seen in astrocytes. Therefore, the cellular conditions that lead to L1 retrotransposition in neurons are also indicative of a commitment to cell death, even in the case of rare events in wild-type animals. By contrast, the cell non-autonomous effects that lead to the death of cells that are nearby to the L1-EGFP-labeled cells only occur in the presence of the hTDP-43 transgenes. Thus, the cell non-autonomous toxicity from cells that exhibit LINE-1 element retrotransposition depends on the presence of pathological TDP-43. It is not clear

why L1-EGFP retrotransposition reporter expression correlates with the onset of motor symptoms in the hTDP43-Q331K animals, but precedes the onset of such neurological effects in the hTDP43-WT animals. It likely derives from differences in cell intrinsic toxicity and cell non-autonomous toxicity of these transgenes and observed differences in cell types in which the L1-EGFP reporter becomes active.

Our finding that many RTEs and ERVs exhibit elevated expression is convergent with the literature that establishes abnormal expression of RTEs and ERVs in response to either TDP-43 or Tau dysfunction in cell culture, in C. elegans, in Drosophila, and in human postmortem tissue [12–15,17–22,34,61–65]. It is worth noting that in postmortem samples from human subjects with ALS, the expression of TEs appears to define only a subset of MC samples, one of three different sub-types based on expression profile [13,30]. While this may reflect heterogeneity in the mechanisms that underlie ALS across subjects, it could also be a result of different disease stages being represented in the tissue samples. Indeed, the transient expression of RTEs and the fleeting presence of neurons and astrocytes that report LINE-1 retrotransposition events in this mouse model may point to a dynamic process that would produce qualitatively different expression profiles across stages of disease progression.

There are a few previous cases where the functional impacts of specific RTEs or ERVs have been studied. Transgenic expression of the HERV-K Env gene alone is sufficient to cause motor dysfunction in a mouse model [20]. HERV-K expression in human neuroblastoma cells is also sufficient to trigger TDP-43 aggregation pathology, indicating that TDP-43 pathology and ERV expression exist in positive feedback [15]. The fly mdg4-ERV (also called gypsy) contributes causally to the toxicity of TDP-43 expressed in glia [12,14]. In the fly model, there is also evidence that mdg4ERV contributes to the cell non-autonomous effects of TDP-43 pathology. Over-expression driven TDP-43 pathology in Drosophila glial cells is sufficient to cause death of nearby neurons, and this cell non-autonomous effect depends on the expression in the glial cells of the mdg4-ERV [14,15].

RNA-seq and ATAC seq experiments from human cortical tissue and RNA-seq from flies with TDP-43 pathology indicate that there is broad up-regulation of RTEs, including not only ERVs but also LINE-1 elements [12,13,19–22,30]. But our findings are the first to provide evidence that LINE-1 elements may also contribute functionally to TDP-43-related neurodegeneration. LINE-1 elements are the only type of RTE for which fully functional copies are present in the human genome [66]. By contrast, there are no fully replication-competent copies of ERVs, although there are several full-length HERV-K proviruses that can express both RNAs and protein products. Thus, while ERVs and LINEs may each have pathological impacts, the mechanisms by which they may cause both cell-intrinsic and cell non-autonomous effects could vary.

There are a number of possible mechanisms by which either LINE-1 or ERV expression could contribute to cellular toxicity. Both LINE-1 elements and ERVs may cause DNA damage, either through retrotransposition or through abortive or unsuccessful retrotransposition events. Indeed, there is growing evidence that DNA damage signaling is at play in neurodegenerative diseases [14,67–70]. There is also evidence that ERV expression can cause protein aggregation, and ERVs can contribute to the intercellular transmission of protein aggregation [15,71]. ERV and LINE-1 RNAs or cDNAs can also trigger the Rig1 and cGAS/STING pathways [72,73], providing a potential source of inflammatory signaling. And finally, both LINE elements and ERVs are potential drivers of cellular senescence [53,74,75].

Each of these mechanisms has the potential to contribute to both cell-intrinsic and cell non-autonomous effects observed in disease progression. Determining the relative contributions of LINE-1 elements and ERVs, as well as the mechanisms by which they impact disease, will be crucial for designing effective clinical interventions. Reverse transcription inhibitors, for example, might be efficacious to prevent effects that depend on cDNA production (already in clinical trials for AD; NCT04552795, NCT04500847, [76–78]), but not effects triggered by LINE-1 or ERV RNAs or proteins. Investigation of the functional impacts from expression of both LINE-1 and ERV elements within an in vivo context will provide the means to decipher the underlying mechanisms by which RTEs contribute to disease pathology.

## Materials and methods

### Ethics statement

All relevant ethical regulations and animal procedures were reviewed and approved by Stonybrook University Division of Laboratory Animal Resources and the Stony Brook University Institutional Animal Care and Use Committee (approval #1133421).

### Mouse husbandry

C57BL/6N mice, B6N.Cg-Tg (Prnp-TARDBP*Q331K)103 Dwc/J Stock # 017933, and B6N.Cg-Tg (Prnp-TARDBP) 96Dwc/J stock # 017907 [33] transgenic mice were purchased from Jackson Laboratories. The human TDP-43 cDNA with or without the Q331K mutation is inserted under the murine prion promoter with an N-terminal myc tag allowing it to be expressed in the central nervous system [33]. The single-copy insert SN1 L1-EGFP animals were a gift from Dr. Jef D. Boeke and Dr. Fred Gage [79]. Briefly, a retrotransposition-competent endogenous mouse LINE-1 element promoter was utilized with codon-optimized mouse L1 ORF1 and ORF2 sequences and and EGFP-based retrotransposition indicator cassette.The EGFP sequence with an intron in the middle is inserted in an inverse fashion under the pCMV promoter in the 3'UTR of the LINE-1 element. Upon transcription, the intron is spliced out. The EGFP is expressed under the pCMV promoter upon reverse transcription and insertion. Thus, the mouse model expresses GFP upon a copy-and-paste event. The hTDP-43-Q331K Tg and the hTDP-43-WT Tg lines were crossed with the L1-EGFP reporter line to create the double transgenic lines. All protocols were approved by IACUC. The animals were housed in the maximum isolation facility and then transferred to the conventional facility prior to the experiments. The cages, bedding and food were sterilized. Food and water were provided ad libitum. A light-dark cycle of 12h was used (7:00 on, 19:00 off). The temperature was maintained between 20–26 °C. For all experiments, mice were used at 1.5, 3, 6, 10, 15, and 17m time points. All cohorts were age-matched and mixed by sex.

### Genotyping

Animals were weaned at 21 days, ear tagged, and a tail clip was taken for genotyping. Genomic DNA extraction was done using homemade buffers. Briefly, 600μl of TNES buffer (Tris-HCL, pH 7.5 = 50ml; 5M NaCl = 80ml; 0.5M EDTA, pH = 8.0 = 20ml; 20% SDS = 30ml; fill up to 1000ml with MilliQ) and 20μl of 10mg/ml proteinase-K were added to each tail clip. This was incubated in a 55 °C water bath overnight. The samples were then neutralized with 180μl, 5M NaCl. After vigorously shaking the samples for a minute, the samples were spun down at 14000 rpm for 20 minutes at room temperature. 600μl of the supernatant was transferred to a fresh tube, and 800μl of ice-cold 100% ethanol was used for precipitating the DNA. The tubes were inverted 10 times and centrifuged for 10 minutes at 14000 rpm. The supernatant was discarded, and the samples were centrifuged again for 2 minutes at 14000 rpm. The remaining supernatant was discarded, and the pellet was dried for 10 minutes and resuspended in 150μl of TE buffer. The genomic DNA was diluted 1:20 in MilliQ water and used for the genotyping PCR (Tables 1–3).

### Behavioral assays

Hindlimb clasping and latency to fall on the accelerated rotarod were used as measures of motor deficits in the mice. Hindlimb clasping is a marker of disease progression in several neurodegenerative disease models, such as ALS, Huntington's disease, and cerebellar ataxias. Mice with lesions in the MC, cerebellum, spinal cord, and basal ganglia show hindlimb clasping [80]. Hindlimb clasping is defined as the retraction of the hindlimbs upon tail suspension. Wild-type animals generally spread out their hindlimbs in anticipation of a surface to mount on. The mice were scored on a severity score of 0–3 [81]. Score 0 for when hindlimbs are consistently splayed. Score 1 for when one hindlimb is partially retracted 50% of the time. Score 2 for when both hindlimbs are partially retracted 50% of the time. Score 3 for when both hindlimbs

**Table 1. Genotyping PCR reaction mixture.**

| | |
|---|---|
| MQ | 4.5 µl |
| Qiagen 2X Master Mix | 7.5 µl |
| 5 uM Forward Primer | 1 µl |
| 5 µM Reverse Primer | 1 µl |
| 1:20 diluted tail genomic DNA | 1 µl |
| Total | 15 µl |

**Table 2. Genotyping primer list.**

| Primer Name | Primer Sequence 5'>3' | Primer Type | PCR Product Size |
|---|---|---|---|
| TDP-43–14748 | AGA GGT GTC CGG CTG GTA G | TDP-43 Transgene Forward | 228 bp |
| TDP-43–14749 | CCT GCA CCA TAA GAA CTT CTC C | TDP-43 Transgene Reverse | |
| oIMR9020 | AAG GGA GCT GCA GTG AAG TA | Internal Positive Control Forward | 297 bp |
| oIMR9021 | CCG AAA ATC TGT GGG AAG TC | Internal Positive Control Reverse | |
| L1-U1 | CAT TTG GGC TGG AGT AGA TT | L1-EGFP Transgene Forward | |
| L1-L1 | AAG GAG GAC GGC AAC AT | L1-EGFP Transgene Reverse | 506 bp (With L1-U1) |
| L1-L2 | AGT GCT TCA GCG GCT AC | L1-EGFP Transgene Reverse | 692 bp (With L1-U1) |

**Table 3. Genotyping PCR thermal cycles.**

| | | |
|---|---|---|
| 94 °C | 3 min | Enzyme activation and initial denaturation |
| 94 °C | 30 sec | 40 cycles |
| 60 °C | 30 sec | |
| 72 °C | 45 sec | |
| 72 °C | 10 min | Final extension |
| 4 °C | ∞ | |

are completely retracted 50% of the time. Testing was always conducted in the afternoon from 15:00–16:00, for all the animals. The accelerated rotarod assay was done in the afternoon from 16:00–18:00 for all the animals. The animals had 1 day of habituation, 1 day of training, and 3 days of testing. Animals were habituated for five minutes on the first day. The mice are placed at a 45° angle onto the rotarod, which is accelerated at a set ramp. The latency to fall from the rotarod is measured. The animals fall onto a soft cushion to avoid injuries from impact. The training consists of three trials, each with a 30-minute inter-trial interval. Trial 1 was done at 0 rpm for 60s. Trials 2 and 3 are done at 0–4rpm for 60s. The testing stage consists of three days, each with three trials. The latency to fall is measured for each animal. Each trial was done at 4–30 rpm for 300s with an inter-trial interval of 30 minutes. The final latency to fall was calculated as an average of all trials over the three days of testing [33].

## RNA-sequencing

We took an n = 3 for the hTDP-43-Q331K Tg, hTDP-43- WT Tg, and non-transgenic littermates. The age groups used for MC tissue collection and RNA extraction were 1.5-, 3-, 6-, 10-, and 15-month animals. All cohorts were mixed-sex groups. Animals were perfused with ice-cold PBS on ice. The MC was dissected on ice and flash-frozen in 100% ethanol and dry ice. The tissue samples were stored at -80 degrees for a few days before being shipped for total RNA extraction and library prep to Azenta. Sequencing was done on the Next-Gen 550 platform. The sequence configuration was Illumina,

2x150bp, with an estimated output of about 350 million raw paired-end reads per lane. Illumina adapters were removed using cutadapt (v4.4) and aligned to the mm39 genome using STAR (v2.7.10b) [82]. Genes and TE were quantified using TE transcripts (v1.2.3) using RefSeq gene annotations (Apr 2023 release) [83] and UCSC RepeatMasker annotations [84]. Differential analysis was performed using DESeq2 (v1.40.2) [85]. Heatmaps were generated with variance stabilizing transformed counts using the NMF R package (v0.27) [86].

## Analysis of enrichment of aging-dependent RTEs in TDP-43-expressing motor cortex

We divided the differentially-expressed RTEs in non-trangenic animals into five arbitrary categories as shown in S5 Table: (i) strongly age-dependent- which showed significant increases in RTEs in at least 3 out of the 4 time points when compared to 15 month-old non-transgenic animals along with no significant decreases in RTEs in the 1.5 month-old animals compared to the 15-month-old animals; (ii) mildly age-dependent- which showed significant increases in RTEs in at least one of the 4 time point comparisons and no significant decreases in RTEs; (iii) 1.5-month peak- which showed significant decreases in RTEs in the 15 month-old non-trangenic animals compared to the 1.5-month-animals but not with the 3-month-old animals; (iv) early expressors- which show significant decreases in RTEs in the 15-month-old animals when compared to the 1.5- and 3-month time points but not in the older time points; and (v) strongly anti-age dependent- significant decreases in RTEs in 15 month-old non-transgenic animals when compared to 1.5-, 3-, and 6-month time points. We observed a significant overlap ($p_{adj}$ = 1.03E-15, odds ratio = 9.9339) in the significantly upregulated RTEs in the 15-month-old hTDP-43-WT and any upregulated RTEs in the 3-month-old hTDP-43-Q331K animals. We decided to use the significantly upregulated RTEs in the 15-month-old hTDP-43-WT animals as the final set of RTEs that are upregulated as a result of TDP-43-related neurodegeneration. We then compared this "RTEs associated with neurodegeneration" list to see if it was enriched in any of the five categories of RTEs expressed in the non-transgenic animals (S5 Table).

## Tissue preparation for IHC

The animals were anesthetized using Isoflurane. The heart was exposed, PBS was injected into the left ventricle, and the right atrium was punctured. The animal was transcardially perfused with PBS for five minutes until the liver and heart cleared. The animal was then perfused with 4% PFA in 1X PBS for five minutes. The animal was decapitated, and the brain was extracted. The brain was transferred to 4% PFA and incubated at 4°C overnight. It was then transferred to 30% sucrose in 1X PBS and incubated on a shaker at 4°C for 2–3 days until the sample sank completely. The brain was then coated with OCT in a brain mold and flash-frozen in dry ice and 100% ethanol. The brain was stored at -80°C until it was cryosectioned. The brain samples were sectioned at a thickness of 50μm and stored as free-floating sections in 1X PBS and stored at 4°C. The samples were stored in 15% glycerol in 1X PBS for long-term storage at -20°C. Standard IHC protocols were used for staining. The sections were blocked in 3% normal donkey serum (NDS- Sigma D9663-10ML) and 0.3% Triton-X 100 in 1X PBS for 3 hours at room temperature on a shaker. The sections were then incubated in the primary antibodies (Table 4) in 1% NDS and 0.1% Triton X- 100 in 1X PBS at 4°C overnight on a shaker. The sections were then washed in 0.1% PBST thrice for 15 minutes on a shaker at room temperature. The secondary antibody was diluted in 1% NDS and 0.1% Triton X- 100 in 1X PBS. The sections were incubated in the secondary antibody dilution for 3 hours at room temperature in the dark with gentle shaking. The sections were then washed three times for fifteen minutes in 0.1% PBST, followed by a final 5-minute wash in PBS. The sections were mounted in Flouromount-G mounting media with or without DAPI from Southern Biotech (Cat# 0100–01 and Cat# 0100–20 respectively), coverslipped, and were allowed to dry for one day before imaging.

## TUNEL staining

We used the TUNEL Assay Kit (Abcam ab66110) to investigate cell death in the brain tissue. The 50μm brain sections, which were perfused, PFA-fixed, and OCT-embedded, were first blocked and permeabilized using 3% normal donkey

**Table 4. Antibody list.**

| Antibody | Company | Catalog Number | Dilution |
|---|---|---|---|
| Rb anti GFP | Invitrogen | A11122 | 1:1000 |
| Chk anti GFAP | Millipore | Ab5541 | 1:1000 |
| Rb anti Iba1 | Fujifilm Wako | 019-19741 | 1:500 |
| Gt anti GFP | Rockland | 600-101-215 | 1:300 |
| Ms IgG1 anti myc tag | Abcam | ab32 | 1:500 |
| Neurotrace- Blue | Invitrogen | N-21479 | 1:250 |
| AF Dk anti Rb 488 | Invitrogen | A-21206 | 1:2000 |
| AF Gt anti Chk 647 | Invitrogen | A-32933 | 1:2000 |
| AF Dk anti Rb 594 | Invitrogen | A-21207 | 1:2000 |
| Rb anti MCM2 | Abcam | Ab4461 | 1:500 |
| Rb X S100B 647 | Abcam | Ab196175 | 1:400 |

serum (NDS - Sigma D9663-10ML) and 0.3% Triton X-100 in 1X PBS. This step was performed for 3 hours at room temperature on a shaker, instead of using proteinase K. Next, the tissue sections were incubated in DNA labeling solution with 0.1% PBST for 48 hours at 4°C while being shaken. When co-labeling with other primary antibodies, such as GFP, the appropriate dilution of the antibody was added together with the DNA labeling solution. After this, the tissue was washed three times with 0.1% PBST for fifteen minutes each on the shaker. Following the washes, the sections were incubated with the anti-BrdU antibody and the secondary antibody against GFP for 1 hour at room temperature in the dark, with shaking. The sections were then washed three times in 0.1% PBST, with each wash lasting 15 minutes, followed by a final 5-minute wash with PBS. Finally, the sections were mounted using Fluoromount-G mounting media with DAPI from Southern Biotech (Cat# 0100-20), coverslipped, and allowed to dry for one day before imaging. For the experiment, control brain tissue was treated with 3000 U/ml of DNase I for 10 minutes at room temperature and served as a positive control.

## Imaging

The Olympus slide scanner was used to image all sections at 10X magnification for quantifying GFP-positive cells. The imaging parameters were kept consistent across different rounds of imaging. A 20µm thick stack was imaged, consisting of 20 optical sections with a 1 µm intervals. The Zeiss LSM 800 confocal microscope was used to study the morphology of various cell types in the GFP-positive cell population using the 20X, 40X and 63X magnifications. Additionally, the Zeiss Airy Scan 63X was used to image the TUNEL staining in the GFP-positive cells. The Zeiss LSM 800 confocal microscope was used at 63X to generate images utilized for the TUNEL distance vs. probability plots.

## Image processing and analysis

We developed a pipeline for the image processing of GFP-positive cell quantification. All images were captured using the same four channels: Neurotrace-blue at 405nm, GFP at 488nm, an empty red channel to correct for background staining, and GFAP at 647nm. Maximum intensity projections were obtained for the green and red channels. The red channel was subtracted from the green channel to eliminate any nonspecific staining. The final image was then thresholded and the number of GFP-positive cells in the MC, striatum, nucleus accumbens, and agranular insular cortex was counted using ImageJ. Boundaries for each section were defined using templates from the Allen Mouse Brain Atlas. The total number of GFP-positive cells was calculated based on the equivalent sections from all the animals.

## Cluster lateralization map

We examined the organization of clusters in 14 sections at identical bregma locations across different animals. Each section was 50 microns thick, with a 150-micron interval between sections. We defined a cluster as a group of cells with LINE-1 retrotransposition events, indicated by GFP expression, which was three standard deviations above the mean GFP-positive cell counts found in the MC of the L1-EGFP littermates. To assess the presence or absence of a cluster, we analyzed total counts across 14 sections. All animals represented on the lateralization map had total GFP counts within the MC that exceeded the cluster threshold. We calculated thresholds for each section based on the mean L1-EGFP cell counts specific to that bregma location. Subsequently, we created a lateralization map that reports the presence, size, and lateralization of clusters. Since we were working with free-floating brain tissue, we utilized the asymmetry between the hemispheres caused by cryosectioning to label the hemispheres as pseudo-left or pseudo-right, and then we tracked the presence of the cluster in each hemisphere. Each row in the map represents an animal, while each column represents a section at the specified bregma location.

## TUNEL: Distance vs. probability plot

To calculate the probability of cell death at various distances from the cell clusters exhibiting LINE-1 retrotransposition events, we conducted imaging from the edge of the GFP-positive clusters or cells. Imaging was performed at 63X on the Zeiss LSM 800 confocal microscope, utilizing a 9 x 3 tile matrix (totaling 27 tiles), which covered a rectangular area of 900 μm x 300 μm. We captured images in the DAPI, GFP, and TUNEL channels. The image was divided into 100μm bins, and the number of TUNEL-positive nuclei was determined in each bin from the raw Z-stack data (Fig 7d). We set an arbitrary threshold of 3 puncta per nucleus for it to be considered TUNEL-positive. We quantified the number of TUNEL-positive nuclei per bin and calculated the probability as follows: (number of TUNEL-positive nuclei/ Total number of nuclei counted).

## Supporting information

**S1 Fig. hTDP-43-Q331K/L1-EGFP Tg mice start showing motor deficits at 3 months, and hTDP-43-WT Tg/L1-EGFP animals do not show deficits until 15 months.** (i-v) Hindlimb Clasping at 1.5, 3, 6, 10, and 15 months. Kruskal- Wallis and Dunn's multiple comparisons test was used with * p ≤ 0.05, ** p < 0.01, *** p < 0.001, **** p < 0.0001. n = 4–9 animals for each group were used. (ii) p = 0.0008 for hTDP-43-Q331K/L1-EGFP vs L1-EGFP, and p = 0.0316 for hTDP-43-Q331K/L1-EGFP vs hTDP-43-WT Tg/L1-EGFP (iii) p = 0.0098 for hTDP-43-Q331K/L1-EGFP vs L1-EGFP, and p = 0.0155 for hTDP-43-Q331K/L1-EGFP vs hTDP-43-WT Tg/L1-EGFP (iv) p = 0.0013 for hTDP-43-Q331K/L1-EGFP vs L1-EGFP (v) p = 0.0202 for hTDP-43-Q331K/L1-EGFP vs L1-EGFP.
(TIF)

**S2 Fig.** (i) Expression of all RTEs across 1.5, 3, 6, 10, and 15 months in hTDP-43-Q331K Tg and hTDP-43-WT Tg mouse MC. (ii) Significantly differentially expressed genes and RTEs at 1.5, 3, 6, 10, and 15 months in hTDP-43-Q331K Tg and hTDP-43-WT Tg mouse MC. N = 3 mixed-sex, age-matched cohorts were used for all genotypes and age groups except 1.5 month, where n = 2 for nTg and n = 4 for hTDP-43-WT Tg were used. See Methods for details regarding the analysis pipeline, including statistical analyses.
(TIF)

**S3 Fig.** (i–x) Volcano plots showing differential expression of genes and RTE in hTDP-43-Q331K and hTDP-43-WT MC compared to non-transgenic littermates at different time points. The grey dotted line indicates –log10padj value of 2. Legend: black = genes, red = RTE.
(TIF)

**S4 Fig. Pathway Analyses and gene ontology analyses of (i-iv) significantly upregulated and (v-vi) downregulated genes in the 3 month-old hTDP-43-Q331K MC compared to non-transgenic littermates.** X- axis represents -log10(Padj) values for each component. In the MC of the 3 month-old hTDP-43-Q331K mice, the differentially expressed transcripts showed enrichment for genes involved in extracellular matrix (ECM) organization. By contrast, there was a depletion of genes associated with neuronal development. Significant differential expression was also identified in genes specific to GABAergic and glutamatergic neuron populations, with some genes being upregulated and some being downregulated, in comparison to the Azimuth annotated reference dataset. In contrast to the hTDP-43-Q331K animals, no significant differences were observed in the total RNA from the MC of 3 month-old hTDP-43-WT mice when compared to their non-transgenic littermates. Figures generated on Enrichr [87–89].
(TIF)

**S5 Fig. Pathway Analyses and gene ontology analyses of significantly upregulated genes in the15 month-old hTDP-43-Q331K MC when compared to non-transgenic littermates.** In the 15 month-old hTDP-43-Q331K MC, differentially expressed transcripts were enriched for ECM receptor interaction, focal adhesion, and PI3K-Akt signaling pathway genes. Additional upregulation was found in genes associated with ECM organization and peptidyl-tyrosine phosphorylation. There was also a significant increase in the expression of genes specific to layer 1–6 astrocytes and GABAergic neurons. X- axis represents -log10(Padj) values for each component. Figures generated on Enrichr [87–89].
(TIF)

**S6 Fig. Pathway Analyses and gene ontology analyses of significantly downregulated genes in the15 month-old hTDP-43-Q331K MC when compared to non-transgenic littermates.** Genes involved in the calcium signaling pathway, glutamatergic synapses, glutamate receptor signaling pathway, and nervous system development were downregulated. This downregulation extended to genes expressed in both glutamatergic and GABAergic neurons. X- axis represents -log10(Padj) values for each component. Figures generated on Enrichr [87–89].
(TIF)

**S7 Fig. Pathway Analyses and gene ontology analyses of (i) significantly upregulated and (ii-v) downregulated genes in the15 month-old hTDP-43-WT MC when compared to non-transgenic littermates.** In the 15 month-old hTDP-43-WT MC, differentially expressed genes were enriched for those associated with GABA-gated chloride ion channel activity and ubiquitin-like protein transferase activity. Additionally, these genes were depleted in gene sets associated with translation, macromolecule biosynthesis, and rRNA metabolism. Genes related to RNA binding, active transmembrane transporter activity, GTPase regulator activity, and ubiquitin ligase inhibitor activity were also significantly downregulated. X- axis represents -log10(Padj) values for each component. Figures generated on Enrichr [87–89].
(TIF)

**S8 Fig. Principal component Analysis showing all significantly expressed RTE in the hTDP-43-Q331K, hTDP-43-WT and non-transgenic littermates.**
(TIF)

**S9 Fig. Principal component Analysis showing all RTE in the hTDP-43-Q331K, hTDP-43-WT and non-transgenic littermates.**
(TIF)

**S10 Fig. Principal component Analysis showing all features in the hTDP-43-Q331K, hTDP-43-WT and non-transgenic littermates.**
(TIF)

**S11 Fig.** i) Heatmap showing expression of RTEs known to be highly expressed in old animals at 1.5-, 3-, 6-, 10- and 15-month time points in the MC total RNA of hTDP-43-Q331K, hTDP-43-WT and non-transgenic littermates. (TIF)

**S12 Fig.** i) Heatmap showing expression of RTEs known to be highly expressed in old animals at 1.5-, 3-, 6-, 10- and 15-month time points in the MC total RNA of hTDP-43-Q331K, hTDP-43-WT and non-transgenic littermates. We note that RTEs that exhibit high expression later in life in control animals are not significantly expressed in the 15-month-old hTDP43-WT animals (padj = 7.31E-04, odds ratio = 0.249). (TIF)

**S13 Fig. Further analysis in the 3-month hTDP-43-Q331K and the 15-month hTDP-43-WT showed a significant differential expression of genes implicated in the cell cycle.** i) Heatmap showing significantly differentially expressed genes involved in the cell cycle at 3 months in the MC total RNA of hTDP-43-Q331K, and non-transgenic littermates. ii) Heatmap showing significantly differentially expressed genes involved in the cell cycle at 15 months in the MC total RNA of hTDP-43-WT and non-transgenic littermates. (TIF)

**S14 Fig.** i) Representative whole brain images from 3- and 6-month-old L1-EGFP, hTDP-43-WT/L1-EGFP and hTDP-43-Q331K/L1-EGFP animals showing LINE-1 retrotransposition clusters in different regions such as the striatum, nucleus accumbens, agranular insular cortex, and piriform cortex. Scalebar = 100µm. (TIF)

**S15 Fig. Representative 10X images from 3- and 6-month-old L1-EGFP, hTDP-43-WT/L1-EGFP and hTDP-43-Q331K/L1-EGFP animals showing LINE-1 retrotransposition clusters in 50 µm sections from the MC.** Scale-bar = 50µm. Bregma images created in Powerpoint. (TIF)

**S16 Fig. LINE-1 retrotransposition events in striatum of hTDP-43-Q331K/L1-EGFP and hTDP-43-WTL1-EGFP mice at 3 months.** (a-e) GFP positive cells/ section in 1.5-, 3-, 6-, 10-, and 15-m, respectively. Images were processed on FIJI-ImageJ and manually counted. Mann- Whitney test was done to compute the significance with * $p \leq 0.05$ and ** $p < 0.01$. p-value for (ii) was 0.0451 and <0.0001, respectively. Mixed cohorts of n = 5–11 animals were used for all genotypes and age groups. hTDP-43-WT/L1-EGFP mice and hTDP-43-Q331K/L1-EGFP mice were maintained as separate colonies and hence have been compared to their littermates L1-EGFP-C1 and L1-EGFP-C2, respectively. (TIF)

**S17 Fig. LINE-1 retrotransposition events in nucleus accumbens of hTDP-43-Q331K/L1-EGFP and hTDP-43-WTL1-EGFP mice at 3 months and 10 months respectively.** (i-v) GFP positive cells/ section in 1.5 m, 3m, 6m, 10m, and 15m, respectively. Images were processed on FIJI- ImageJ and manually counted. Mann- Whitney test unpaired t-test was done to compute the significance with * $p \leq 0.05$ and ** $p < 0.01$. p-value for (ii) was 0.0220 and for (iv) was 0.0098 respectively. Mixed cohorts of n = 5–11 animals were used for all genotypes and age groups. hTDP-43-WT/L1-EGFP mice and hTDP-43-Q331K/L1-EGFP mice were maintained as separate colonies and hence have been compared to their littermates L1-EGFP-C1 and L1-EGFP-C2, respectively. (TIF)

**S18 Fig. LINE-1 retrotransposition events in agranular insular cortex of hTDP-43-Q331K/L1-EGFP mice at 3 months and 6 months respectively.** (i-v) GFP positive cells/ section in 1.5 m, 3m, 6m, 10m, and 15m, respectively. Images were processed on FIJI- ImageJ and manually counted. Mann- Whitney test unpaired t-test was done to compute the significance with * $p \leq 0.05$ and ** $p < 0.01$. p-value for (ii) was 0.0186 and for (iii) was 0.0375. Mixed cohorts of n = 5–11 animals were used for all genotypes and age groups. hTDP-43-WT/L1-EGFP mice and hTDP-43-Q331K/

L1-EGFP mice were maintained as separate colonies and hence have been compared to their littermates L1-EGFP-C1 and L1-EGFP-C2, respectively.
(TIF)

**S1 Table. Fraction of animals with clusters at different time points in different genotypes.**
(XLSX)

**S2 Table. Motor cortex RNA- sequencing mapping_QC.**
(XLSX)

**S3 Table. TDP43-Q331KvNonTg_diffExp.**
(XLSX)

**S4 Table. TDP43-WTvNonTg_diffExp.**
(XLSX)

**S5 Table. Analysis of enrichment of aging-dependent RTEs in TDP43-expressing motor cortex.**
(XLSX)

**S6 Table. TDP43-WT v NonTg_diffExp.**
(XLSX)

**S7 Table. TDP43-Q331K v NonTg_diffExp.**
(XLSX)

**S8 Table. TDP43-Q331K v TDP43-WT_diffExp.**
(XLSX)

**S9 Table. Age_increase_TE.**
(XLSX)

**S10 Table. High in young_TE.**
(XLSX)

**S11 Table. Mouse_activeL1.**
(XLSX)

**S12 Table. GFP counts per section in different regions and time points.**
(XLSX)

**S13 Table. Lateralization of Clusters at 3 and 6 months in the motor cortex.**
(XLSX)

**S14 Table. Cell type specificity of motor cortex clusters.**
(XLSX)

**S15 Table. TUNEL staining_distance vs percentage plot raw counts.**
(XLSX)

## Acknowledgments

We wish to thank current and former members of both the Sher and Dubnau labs for critical discussions and advice. We wish to thank Crystal Huang and Michael Petagna for laboratory assistance. We wish to thank Dr. Tim Tully for advice and consultation on statistical analyses. We wish to thank Wendy Ackmentin for her invaluable help on all aspects of running lab facilities.

## Author contributions

**Conceptualization:** Shreevidya Korada, Josh Dubnau, Roger B. Sher.

**Data curation:** Roger B. Sher.

**Formal analysis:** Shreevidya Korada, Oliver H. Tam, Molly Gale Hammell, Roger B. Sher.

**Funding acquisition:** Josh Dubnau, Roger B. Sher.

**Investigation:** Shreevidya Korada, Hunter C. Greco, Roger B. Sher.

**Methodology:** Shreevidya Korada, Oliver H. Tam, Molly Gale Hammell, Roger B. Sher.

**Project administration:** Roger B. Sher.

**Resources:** Josh Dubnau, Roger B. Sher.

**Software:** Roger B. Sher.

**Supervision:** Josh Dubnau, Roger B. Sher.

**Validation:** Shreevidya Korada, Oliver H. Tam, Molly Gale Hammell, Josh Dubnau, Roger B. Sher.

**Visualization:** Shreevidya Korada, Oliver H. Tam, Roger B. Sher.

**Writing – original draft:** Shreevidya Korada, Josh Dubnau, Roger B. Sher.

**Writing – review & editing:** Shreevidya Korada, Oliver H. Tam, Molly Gale Hammell, Josh Dubnau, Roger B. Sher.

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
