## [Decision Letter · Decision Letter 0]

31 Aug 2025

PGENETICS-D-25-00724

LINE-1 replication in a mouse TDP-43 model of neurodegeneration marks motor cortex neurons for cell-intrinsic and non-cell autonomous programmed cell death

PLOS Genetics

Dear Dr. Sher,

Thank you for submitting your manuscript to PLOS Genetics. After careful consideration, we feel that it has merit but does not fully meet PLOS Genetics's publication criteria as it currently stands. Therefore, we invite you to submit a revised version of the manuscript that addresses the points raised during the review process.

Please submit your revised manuscript within 60 days Oct 30 2025 11:59PM. If you will need more time than this to complete your revisions, please reply to this message or contact the journal office at plosgenetics@plos.org. Please include the following items when submitting your revised manuscript:

We look forward to receiving your revised manuscript.

Kind regards,

J. Nicholas Cochran

Guest Editor

PLOS Genetics

Hua Tang

Section Editor

PLOS Genetics

Aimée Dudley

Editor-in-Chief

PLOS Genetics

Anne Goriely

Editor-in-Chief

PLOS Genetics

**Journal Requirements:**

3) Some material included in your submission may be copyrighted. According to PLOSu2019s copyright policy, authors who use figures or other material (e.g., graphics, clipart, maps) from another author or copyright holder must demonstrate or obtain permission to publish this material under the Creative Commons Attribution 4.0 International (CC BY 4.0) License used by PLOS journals. Please closely review the details of PLOSu2019s copyright requirements here: PLOS Licenses and Copyright. If you need to request permissions from a copyright holder, you may use PLOS's Copyright Content Permission form.

Potential Copyright Issues:

i) Please confirm (a) that you are the photographer of 1A, or (b) provide written permission from the photographer to publish the photo(s) under our CC BY 4.0 license.

ii) Figures 1, 4a, 6a, and S3b. Please confirm whether you drew the images / clip-art within the figure panels by hand. If you did not draw the images, please provide (a) a link to the source of the images or icons and their license / terms of use; or (b) written permission from the copyright holder to publish the images or icons under our CC BY 4.0 license. Alternatively, you may replace the images with open source alternatives. See these open source resources you may use to replace images / clip-art:

4) Please amend your detailed Financial Disclosure statement. This is published with the article. It must therefore be completed in full sentences and contain the exact wording you wish to be published.

2) If any authors received a salary from any of your funders, please state which authors and which funders.

5) Please ensure that the funders and grant numbers match between the Financial Disclosure field and the Funding Information tab in your submission form. Note that the funders must be provided in the same order in both places as well. Currently, the order of the funders are not the same in both places.

**Reviewers' comments:**

Reviewer's Responses to Questions

**Comments to the Authors:**

**Please note that one review is uploaded as an attachment.**

Reviewer #1: The review is uploaded as an attachment.

Reviewer #2: Korada and colleagues investigate the role of TDP-43 in repressing retrotranspon elements (RTEs) using an established two mouse models of TDP-43 pathology, the mild overexpression of wildtype or Q331K human TDP-43. They show that the Q331K mouse has early motor deficits compared to a later deficit in the WT. They cross the mice to a LINE1 GFP reporter line to observe LINE1 re-integration in the motor cortex. They show that this LINE1 integration is enhanced at particular time points in the TDP-43 mutant lines, and that this is accompanied by markers of cell death in LINE1-positive neurons, along with nearby cells.

This is a very interesting paper for the field, showing that TDP-43 is associated with LINE de-repression and suggesting that this de-repression is killing both neurons and neighboring cells. Overall, the paper is well written and the methods are clearly described. I have few major comments, but would suggest that the figures are improved to be more legible and consistent across panels.

Major comment: are the authors planning to look at TDP-43 dependent splicing changes in their RNA-seq data? It would be useful to show the accumulation of both loss-of-function (cryptic exons) and gain of function (skiptic exons) at different time points to see whether these splicing changes co-occur with the RTE changes observed. There is a database of mouse TDP-43 splicing events in Fratta et al, 2018 (PMID: 29764981) that could be queried.

Reviewer #3: This is a very nice, compact paper which makes some important discoveries that advance the field as a whole. The transient upregulation of RTEs is a significant finding. The correlation of the upregulation with onset of neurological phenotypes in the two models (WT and mutant TDP) is a significant finding. The death of neurons by apoptosis, and the “action at a distance” (non-cell autonomous death of neighboring cells) are also significant findings, and help to explain previous findings on RTE expression in human post-mortem samples. As a whole, the data are of sufficient quality to support these conclusions, and this warrants the publication of this study.

Major issues

While the apparent increase of L1 retrotransposition, as evidenced by the L1-EGFP reporter, is a very nice touch, given the results obtained, it also injects a certain degree of uncertainty on what’s actually going on.

1) GFP-positive “cell clusters” are seen at 3 and 6 months but not later. This correlates well with the wave of endogenous RTE expression seen in the mutant-TDP mice but not at all in the WT-TDP mice. I’m struggling to find an explanation for this. The authors should provide some speculations on this.

2) Relevant to the above, the L1-EGFP line is poorly defined (I looked up the Gage paper). It was made by pronuclear injection, so likely contains many copies of the reporter (at one or several genomic loci). Is anything known about this? These factors probably affect its expression and regulation. Also, how many times has the line been backcrossed to C57? The TDP models from JAX are on a C57BN background. Are the experiments in this paper done on a mixed background?

3) The GFP signal tracks successful retrotransposition but not expression of the reporter itself. The L1-EGFP is human, and thus would be expressed either from its own (quite weak) promoter (inside the 5’ UTR) or from a cellular promoter at the site of its insertion. I see no reason why the human L1 5’UTR would be regulated in concert with endogenous mouse L1s. Given that the authors note that the upregulation of endogenous RTEs was “similar across both evolutionarily recent and more ancient transposable elements”, one possibility is that the upregulation is rather nonspecific and reflects a large-scale chromatin relaxation. It is possible that the transgene integrated in a heterochromatic region and thus could also ride this wave. In any case, it’s easy to monitor the expression of the L1-EGFP reporter (it’s human) by qRT-PCR, and this should be done. An alternative explanation could be that the reporter is on all the time, but that the cellular environment becomes supportive of productive retrotransposition at the observed times. It is important to address these issues because of the discrepancy noted in 1) above.

4) Currently, endogenous RTE expression is monitored by RNA-seq. This is limiting for two reasons: 1) well known issues of deconvoluting expression of repetitive sequences (and assigning functional vs. non-functional L1s), 2) does not provide single-cell resolution, while much of the other data in this paper do. There is a good antibody to mouse L1 ORF1 (validated by Alex Bortvin, Abcam AB216324), and it would be easy to perform and include those experiments on the specimens already in hand.

Minor issues

1) The authors keep referring to “clusters”, defined as “large, spatially defined groups of L1-EGFP-labeled cells”. The manner in which they are scored is described verbally (which is clear), but it would be very helpful to have some visuals on this. For example, in some of the images presented, draw a dashed line around what is being referred to as a “cluster”.

2) Row designations in Fig. 2a, b are missing.

3) Is the lifespan of mutant-TDP Tg mice known? They are missing from the 17 month timepoint.

4) Fig. 3: not clear what is being plotted here. Each dot corresponds to one mouse, and the position on the Y axis is the average number of GFP positive cells in that mouse? How were the violin outlines computed?

5) Sentence line 709-711, meaning is not clear. What is meant by any upregulated RTE? Both significantly and non-significantly upregulated RTEs? Was there no significant overlap between significantly upregulated in WT hTDP and significantly upregulated in Q331K hTDP?

6) Sentence on line 713-715: “We then compared . . . “ Where is the result of this comparison? Something seems to be missing.

7) The authors keep referring to “replication” of L1, as evidenced by the L1-EGFP reporter. I would prefer to use “retrotransposition” – I believe it is a more descriptive and accurate term.

**Have all data underlying the figures and results presented in the manuscript been provided?**

Reviewer #1: None

Reviewer #2: Yes

Reviewer #3: Yes

PLOS authors have the option to publish the peer review history of their article (what does this mean? ). If published, this will include your full peer review and any attached files.

**Do you want your identity to be public for this peer review?** For information about this choice, including consent withdrawal, please see our Privacy Policy .

Reviewer #1: No

Reviewer #2: No

Reviewer #3: No

**Figure resubmission:**
---

## [Decision Letter · Decision Letter 1]

19 Dec 2025

Dear Dr Sher,

We are pleased to inform you that your manuscript entitled "LINE-1 retrotransposition in a mouse TDP-43 model of neurodegeneration marks motor cortex neurons for cell-intrinsic and cell non-autonomous programmed cell death" has been editorially accepted in principle for publication in PLOS Genetics. Congratulations! Please address the final minor points by reviewer 3 along with requested editorial formatting (see below).

Yours sincerely,

J. Nicholas Cochran

Guest Editor

PLOS Genetics

Hua Tang

Section Editor

PLOS Genetics

Aimée Dudley

Editor-in-Chief

PLOS Genetics

Anne Goriely

Editor-in-Chief

PLOS Genetics

BlueSky: @plos.bsky.social

Comments from the reviewers (if applicable):

Reviewer's Responses to Questions

**Comments to the Authors:**

Reviewer #1: The authors have addressed the reviewers' questions and concerns. Each point has been clearly responded to, and the corresponding revisions have been appropriately incorporated into the manuscript. The overall of the manuscript is clearer after the corrections and revisions.

Reviewer #2: I am satisfied that the authors have addressed my comments.

Reviewer #3: Rev. 3, point 1): Please include your thoughts and and interpretations in the Discussion. Speculative is fine, this is what discussions are for.

Point 3): please include your response (or a shortened version of it) in the discussion.

**Have all data underlying the figures and results presented in the manuscript been provided?**

Reviewer #1: Yes

Reviewer #2: Yes

Reviewer #3: Yes

PLOS authors have the option to publish the peer review history of their article (what does this mean? ). If published, this will include your full peer review and any attached files.

**Do you want your identity to be public for this peer review?** For information about this choice, including consent withdrawal, please see our Privacy Policy .

Reviewer #1: No

Reviewer #2: No

Reviewer #3: **Yes: ** John Sedivy

**Data Deposition**

http://datadryad.org/submit?journalID=pgenetics&manu=PGENETICS-D-25-00724R1

**Press Queries**

---

## [Editor Report · Acceptance letter]

PGENETICS-D-25-00724R1

LINE-1 retrotransposition in a mouse TDP-43 model of neurodegeneration marks motor cortex neurons for cell-intrinsic and cell non-autonomous programmed cell death

Dear Dr Sher,

We are pleased to inform you that your manuscript entitled "LINE-1 retrotransposition in a mouse TDP-43 model of neurodegeneration marks motor cortex neurons for cell-intrinsic and cell non-autonomous programmed cell death" has been formally accepted for publication in PLOS Genetics! Your manuscript is now with our production department and you will be notified of the publication date in due course.

With kind regards,

Anita Estes

PLOS Genetics

On behalf of:
